



**Simulation of radon-222 with the GEOS-Chem global model:**

**Emissions, seasonality, and convective transport**

Bo Zhang[1], Hongyu Liu[1], James H. Crawford[2], Gao Chen[2], T. Duncan Fairlie[2], Scott Chambers[3],

Chang-Hee Kang[4], Alastair G. Williams[3], Kai Zhang[5], David B. Considine[6], Melissa

P. Sulprizio[7], and Robert M. Yantosca[7]

[1]National Institute of Aerospace, Hampton, Virginia, USA

[2]NASA Langley Research Center, Hampton, Virginia, USA

[3]Australian Nuclear Science and Technology Organization, Kirrawee, New South Wales, Australia

[4]Jeju National University, Jeju, Republic of Korea

[5]Pacific Northwest National Laboratory, Richland, Washington, USA

[6]NASA Headquarters, Washington, D.C., USA

[7]Harvard University, Cambridge, Massachusetts, USA

Manuscript submitted to ACPD, July 2020

*Correspondence to*: Hongyu Liu (hongyu.liu-1@nasa.gov)

**Abstract.** Radon-222 ($^{222}$Rn) is a short-lived radioactive gas naturally emitted from land surfaces, and has

long been used to assess convective transport in atmospheric models. In this study, we simulate $^{222}$Rn using

the GEOS-Chem chemical transport model to improve our understanding of $^{222}$Rn emissions and surface

concentration seasonality, and characterize convective transport associated with two Goddard Earth

Observing System (GEOS) meteorological products, MERRA and GEOS-FP. We evaluate four global

$^{222}$Rn emission scenarios by comparing model results with observations at 51 surface sites. The default

emission scenario in GEOS-Chem yields a moderate agreement with surface observations globally (< 70%





data within a factor of 2) and a large underestimate of winter surface $^{222}$Rn concentrations at Northern

Hemisphere mid- and high-latitudes due to an oversimplified formulation of $^{222}$Rn emission fluxes (1 *atom*

*cm$^{-2}$ s$^{-1}$* over land with a reduction by a factor of 3 under freezing conditions). We compose a new global

$^{222}$Rn emission scenario based on Zhang et al. (2011) and demonstrate its potential to improve simulated

surface $^{222}$Rn concentrations and seasonality. The regional components of this scenario include spatially

and temporally varying emission fluxes derived from previous measurements of soil radium content and

soil exhalation models, which are key factors in determining $^{222}$Rn emission flux rates. However, large

model underestimates of surface $^{222}$Rn concentrations still exist in Asia, suggesting unusually high regional

$^{222}$Rn emissions. We therefore propose a conservative up-scaling factor of 1.2 for $^{222}$Rn emission fluxes in

China, which was also constrained by observed deposition fluxes of $^{210}$Pb (a progeny of $^{222}$Rn). With this

modification, the model shows better agreement with observations in Europe and North America (>80%

data within a factor of 2), and reasonable agreement in Asia (close to 70%). Further constraints on $^{222}$Rn

emissions would require additional concentration and emission flux observations in the central U.S.,

Canada, Africa, and Asia. We also compare and assess convective transport in model simulations driven by

MERRA and GEOS-FP using observed $^{222}$Rn vertical profiles in northern mid-latitude summer, and from

three short-term airborne campaigns. While simulations with both GEOS products are able to capture the

observed vertical gradient of $^{222}$Rn concentrations in the lower troposphere (0-4 km), neither correctly

represents the level of convective detrainment, resulting in biases in the middle and upper troposphere.

Compared with GEOS-FP, MERRA leads to stronger convective transport of $^{222}$Rn, which is partially

compensated by its weaker large-scale vertical advection, resulting in similar global vertical distributions of

$^{222}$Rn concentrations between the two simulations. This has important implications for using chemical

transport models to interpret the transport of other trace species when these GEOS products are used as

driving meteorology.



# 1 Introduction

A reasonable representation of transport processes in global models is critical to properly simulate tropospheric trace gases and aerosols. However, convective transport and boundary-layer turbulent mixing occur at sub-grid scales and are usually parameterized, unavoidably introducing transport biases. Radon-222 ($^{222}$Rn, half-life 3.8 days), an atmospheric radionuclide, is an excellent tracer for assessing these biases due to its relatively well-constrained sources and fairly simple sink pathway (radioactive decay) in the atmosphere (Jacob et al., 1997; Liu et al., 1984). In this work, we evaluate and improve the simulation of $^{222}$Rn in a global chemical transport model (GEOS-Chem CTM) and assess the role of convective transport in shaping $^{222}$Rn vertical distributions.

Radon-222 is an inert gas ubiquitously produced in soils and rocks by radioactive decay of radium ($^{226}$Ra). Shortly after $^{222}$Rn emanates to the atmosphere it decays to $^{210}$Pb (half-life 22.3 years). Wet and dry depositions of $^{222}$Rn are negligible due to its inert nature. The spatial distribution of $^{222}$Rn is therefore strongly affected by convective and synoptic-scale transport. Numerous studies have used $^{222}$Rn to evaluate model transport processes, such as boundary-layer structure and stability, vertical motion and mixing, and convection. For instance, Liu et al. (1984) derived seasonal vertical eddy diffusion coefficients using observed vertical profiles of tropospheric $^{222}$Rn concentrations. Allen et al. (1996) used $^{222}$Rn profile measurements to evaluate moist convection in their model and showed that deep convection from the boundary layer to the upper troposphere facilitates the formation of a "C-like" $^{222}$Rn vertical profile. Considine et al. (2005) used $^{222}$Rn and $^{210}$Pb measurements to examine the roles of convective transport in three different meteorological data sets. Zhang et al. (2008) tested two widely used convection schemes, Zhang-McFarlane-Hack (Hack, 1994; Zhang and McFarlane, 1995) and Tiedtke-Nordeng (Nordeng, 1994), in a global circulation model against observed $^{222}$Rn profiles. Although model results with both schemes showed similarly reasonable estimates of surface $^{222}$Rn concentrations, some degree of discrepancies were found in the middle and upper troposphere. $^{222}$Rn has also been used as an indicator of continental




influences on remote marine regions (Balkanski et al., 1992; Chambers et al., 2013, 2018). Model inter-comparison of simulated $^{222}$Rn distributions has been an efficient approach to compare and contrast transport characteristics with respect to boundary-layer turbulent mixing and convection (Genthon and Armengaud, 1995; Jacob et al., 1997).

$^{222}$Rn emission fluxes have been estimated based on: (1) direct measurements, usually by assuming a linear increase of $^{222}$Rn in a chamber placed on soil, and (2) indirect estimates, through an integration of $^{222}$Rn profiles by assuming a local balance between $^{222}$Rn emission and decay. Using both approaches, Wilkening et al. (1972, 1975) derived an estimate of global mean $^{222}$Rn emission fluxes (0.75 *atom cm$^{-2}$ s$^{-1}$* over land). Turekian et al. (1977) later suggested this global mean flux rate was likely an underestimate due

to the assumption of a local steady state. By also considering one-dimensional longitudinal transport, Turekian et al. (1977) recommended a higher global mean flux of 1.2 *atom cm$^{-2}$ s$^{-1}$*, which led to a better agreement with observed $^{210}$Pb deposition fluxes across various latitudes. More recently, a mean global emission flux of 1.0 *atom cm$^{-2}$ s$^{-1}$* was considered more accurate, and has been used uniformly over land as a standard configuration (Balkanski et al., 1993). $^{222}$Rn fluxes from water surface are a few orders of

magnitude lower and can be neglected compared with emissions over land (Wilkening and Clements, 1975). To date, most global models have used a globally uniform $^{222}$Rn emission flux of 1.0 *atom cm$^{-2}$ s$^{-1}$* with modifications in high latitudes and for freezing soil temperatures.

Quantification of regional and temporal $^{222}$Rn emission variations has been extended to broader areas and improved by new measurement techniques and modeling approaches. Observations have indicated that

local $^{222}$Rn emission fluxes vary extensively with surface texture, soil moisture, radium content, ice coverage, and freezing condition (Martell, 1985; Turekian et al., 1977). The increasing availability of observational data inspired studies to quantify regional and temporal emission variations. Based on a large collection of global observations, Conen and Robertson (2002) proposed a linearly decreasing gradient in the Northern Hemisphere, from 1 atom cm$^{-2}$ s$^{-1}$ at 30°N to 0.2 *atom cm$^{-2}$ s$^{-1}$* at 70°N. Regional and global

$^{222}$Rn emission flux datasets at fine resolution have also been developed based on models of gas diffusion





in porous media; this was facilitated by increasingly available soil parameters from meteorological models and assimilation (or reanalysis) datasets. Genthon and Armengaud (1995) introduced soil parameters into a global GCM to formulate online soil-atmosphere exchange of $^{222}$Rn, which assisted in capturing rapid fluctuations of surface $^{222}$Rn concentrations over freezing surfaces. Zhuo et al. (2008) compiled radium

content information from over a thousand sites in China and constructed a high spatial resolution emission map over the country. Hirao et al. (2010) constructed a decade-long global $^{222}$Rn emission record based on additional considerations about surface texture. Due to the availability of extensive  measurements of $^{222}$Rn emission fluxes and surface concentrations, Europe has the finest resolution emission inventory of up to $0.083° \times 0.083°$ with variability in  regional and temporal emissions (Karstens et al., 2015; López-Coto et

al., 2013; Szegvary et al., 2009). Such variability is missing in the current GEOS-Chem standard model, which limits the use of $^{222}$Rn as a tracer to evaluate model transport processes, not to mention that $^{222}$Rn emission and distributions directly affect the production of its progeny $^{210}$Pb, a useful tracer for testing aerosol transport and wet deposition (Liu et al., 2001; Considine et al., 2005).

GEOS-Chem is driven by assimilated meteorological data sets archived from the Goddard Earth

Observing System (GEOS) of the NASA Global Modeling and Assimilation Office (GMAO). Changes in the model dynamics often occur as the GEOS model evolves, which in turn affect the characteristics of transport of chemical species in GEOS-Chem. In an evaluation with satellite observed carbon monoxide in the upper troposphere and lower stratosphere (UTLS), Liu et al. (2013) reported that less carbon monoxide was lofted to the UTLS when GEOS-Chem was driven by the GEOS-5 assimilated data due to insufficient

vertical transport compared with GEOS-4. Downward transport from stratosphere to troposphere was previously found to be substantially overestimated in CTMs driven by GEOS-1 compared with  GEOS-4 (Liu et al., 2016). In a similar manner, evaluation of $^{222}$Rn simulations with observations will help characterize convective transport and its uncertainty in the GEOS series.

In this paper, we assess and improve the simulation of $^{222}$Rn as a model utility to test convective

transport in GEOS-Chem. We  incorporate  into  the  model  recently  published  global  $^{222}$Rn  emission



scenarios. We conduct model simulations with varying emission configurations and provide insights into potential biases in regional and seasonal emissions through evaluations against observed surface $^{222}$Rn concentrations and vertical profiles. We also present the apparent changes in simulated $^{222}$Rn vertical distributions as the driving meteorology switches between the Modern-Era Retrospective analysis for

Research and Applications (MERRA) and GEOS Forward Processing (GEOS-FP), with a specific focus on the role of convection.

The rest of this paper is organized as follows. Section 2 describes the GEOS-Chem model, four $^{222}$Rn emission scenarios, model simulations, and observational data sets used in this work. Section 3 evaluates the four different $^{222}$Rn emission scenarios by comparing simulated $^{222}$Rn with surface measurements.

Section 4 discusses potentially excessive $^{222}$Rn emissions in Asia. Section 5 examines simulated surface $^{222}$Rn seasonality at selected sites. Section 6 assesses convective transport in the model and compares the role of convective transport in MERRA and GEOS-FP in the $^{222}$Rn vertical distribution.

## 2 Model and data

### 2.1 GEOS-Chem

GEOS-Chem (http://www.geos-chem.org) is a global 3-D CTM of atmospheric composition with aerosol-chemistry interactions in both the troposphere and stratosphere, driven by GEOS assimilated meteorological fields from the NASA GMAO (e.g., Bey et al., 2001; Park et al., 2004; Eastham et al., 2014; ). The model uses the TPCORE advection algorithm of Lin and Rood (1996). Convective transport is

calculated using archived convective mass fluxes (Wu et al., 2007). Boundary-layer mixing is based on the non-local scheme implemented by Lin and McElroy (2010). In this study, we use two different GEOS products (MERRA and GEOS-FP) to drive the model simulations. MERRA is a 30-year reanalysis product based on GEOS-5.2.0 (Rienecker et al., 2011). Its native resolution is 0.667° longitude by 0.5° latitude, with 72 vertical layers from the surface up to 80 *km*. GEOS-FP is the current operational product of GEOS-

5.7 (and after) using an analysis developed jointly with NOAA's National Centers for Environmental



Prediction (NCEP). It has a native resolution of 0.3125° longitude by 0.25° latitude with the same vertical grids as MERRA. In all simulations for this study we used the MERRA and GEOS-FP fields regridded to 2.5° longitude by 2° latitude with vertical layers reduced to 47 levels. The meteorological archives have temporal resolutions of 3 h for 3-D fields and 1 h for 2-D fields. MERRA and GEOS-FP use similar model

schemes for fundamental dynamical and physical processes. They both use the modified Relaxed Arakawa-Schubert scheme for convection (Moorthi and Suarez, 1992) and a combined turbulence parameterization based on Lock et al. (2000) and Louis et al. (1981). Compared with MERRA, GEOS-FP made a few adjustments including, but not limited to, increasing re-evaporation in precipitation and adjusting the balance between local and non-local turbulent diffusion, with the former resulting in considerable increase

in water vapor in the tropical troposphere (Molod et al., 2012). MERRA-2, which shows improved climate over MERRA (Molod et al., 2015), is not used here since it was not ready to drive our model when this study was started.

GEOS-Chem      (v11-01f,      http://wiki.seas.harvard.edu/geos-chem/index.php/GEOS-Chem_v11-01) includes a radionuclide ($^{222}$Rn-$^{210}$Pb-$^{7}$Be) simulation option, which runs independently from the full

oxidant-aerosol chemistry simulation. The radionuclide tracers have been used to evaluate chemical transport (Jacob et al., 1997; Yu et al., 2018) and wet deposition processes (Liu et al., 2001, 2004) in the model. The simulation of $^{222}$Rn includes emissions, transport (advection, convection, boundary layer mixing), and radioactive decay. Wet and dry deposition of $^{222}$Rn are neglected in the model.

**2.2 Rn emission scenarios**

The standard version of GEOS-Chem uses the $^{222}$Rn emission scenario of Jacob et al. (1997) (hereafter referred to as JA97). The World Climate Research Program (WCRP) Cambridge Workshop of 1995 (Rasch et al., 2000) previously used JA97 to compare $^{210}$Pb deposition processes in multiple atmospheric models. JA97 was developed using the estimated global mean $^{222}$Rn fluxes of Turekian et al. (1977) and only considered emission variations for a few broad latitude bands. The $^{222}$Rn emission fluxes in JA97 are

uniformly set to be 1 *atom cm$^{-2}$ s$^{-1}$* over land between 60°N – 60°S, 0.005 *atom cm$^{-2}$ s$^{-1}$* between 60°N –



70°N and 60°S – 70°S, and zero poleward of 70°N/S. Emission fluxes over lakes and oceans are set to 0.005 *atom cm⁻² s⁻¹*. Emissions are reduced by a factor of 3 when surface temperature is below 0°C on account of the depressed exhalation of $^{222}$Rn under freezing conditions. Such a temperature-dependent reduction can lead to underestimated emissions in winter because soils may not be totally frozen when

temperature falls below 0°C for only a short period of time. The overall uncertainty of the JA97 emission was estimated to be within 25% globally (Jacob et al., 1997).

A few $^{222}$Rn emission scenarios were published after Jacob et al. (1997). Conen and Robertson (2002) proposed a $^{222}$Rn emission scenario having a uniform $^{222}$Rn flux of 1 *atom cm⁻² s⁻¹* from the continental surface in the Southern Hemisphere and tropics and a linear decreasing trend from 1 *atom cm⁻² s⁻¹* at 30°N

to 0.2 *atom cm⁻² s⁻¹* at 70°N. This decreasing trend towards high latitudes was supported by experimental results showing a decrease of $^{222}$Rn fluxes in the local soil with higher water table and organic portion. This proposed latitudinal gradient was found to be in a good agreement with an estimated value based on multi-year observations at a few Asian sites (Williams et al., 2009). However, this emission scenario is not used in this work because it does not include regional variations other than the linear latitudinal gradient.

Schery and Wasiolek (1998, hereafter SW98) published the first global $^{222}$Rn emission inventory that included detailed regional and seasonal variations on a monthly basis (at 1° longitude by 1° latitude resolution). The emission flux in SW98 is formulated by using a theoretical diffusion model of porous soil with controlling factors of soil radium content, soil moisture, and surface temperature. The estimated annual global average $^{222}$Rn emission flux was 1.63 ± 0.43 *atom cm⁻² s⁻¹*, higher than the widely used JA97

constant value of 1 *atom cm⁻² s⁻¹*. Global $^{222}$Rn emissions in SW98 exhibited regional variations of a factor of 3 and seasonal variations of a factor of 2. The dominant factor in determining the regional variations in $^{222}$Rn fluxes was found to be the soil radium concentrations according to Schery and Wasiolek (1998). Emission fluxes in the U.S. and China feature more detailed regional variations because soil radium concentrations in these countries were incorporated. However, it was suggested that SW98 overestimated

the global average $^{222}$Rn flux (Koch et al., 2006; Zhang et al., 2011). When using the SW98 emissions in a



global model simulation of $^{210}$Pb, Koch et al. (2006) found it necessary to reduce the emissions by half to improve the excessive $^{210}$Pb concentrations in their model. In this study, we reduce the emission fluxes of SW98 globally by a factor of 1.6, as recommended by Zhang et al. (2011).

Zhang et al. (2011, hereafter ZK11) compiled a new global $^{222}$Rn emission inventory based on a
combination of SW98 (with a global reduction factor of 1.6) and recently published $^{222}$Rn flux measurements in Europe and the U.S. (Szegvary et al., 2007), China (Zhuo et al., 2008), Australia (Griffiths et al., 2010), and  oceanic regions (Schery and Huang, 2004). In ZK11, $^{222}$Rn emissions in Europe were derived from a demonstrated linear relationship between terrestrial gamma dose rate and $^{222}$Rn emissions (Szegvary et al., 2007). The relationship allows a convenient calculation of regional $^{222}$Rn
emissions for places where automatic measurements of gamma dose rate have been established, e.g., in Europe (Szegvary et al., 2009). This $^{222}$Rn emission inventory for Europe has recently been updated (Karstens et al., 2015; López-Coto et al., 2013) with further detailed information on soil and surface roughness and minor modifications about handling $^{222}$Rn transport in porous media. A high-resolution (25 km × 25 km) $^{222}$Rn emission map for China was included in ZK11 based on work by Zhuo et al. (2008),
who estimated the nation-wide emissions according to measurements of radium content in surface soil at 1099 sites in China. The oceanic emission flux used was 0.00182 *atom cm$^{-2}$ s$^{-1}$*, derived by Schery and Huang (2004) with a gas transfer model, significantly lower than typical $^{222}$Rn emissions over land.

In this study, we modify ZK11 to a customized $^{222}$Rn emission scenario (hereafter referred to as ZKC) and constrain the inventory with observations of surface $^{222}$Rn concentrations. This customized emission
scenario adopts ZK11 for most areas except for North America, where the SW98 emission fluxes are used with a reduction factor of 1.6, following previous model studies (Koch et al., 2006; Zhang et al., 2011). We also increase the emission over China by a factor of 1.2 all year round due to potentially excessive $^{222}$Rn emission there, which will be discussed in detail in Section 3.2. The emission enhancement factor is only tentative due to very few surface $^{222}$Rn measurements available in western China and a lack of seasonality
in the measurements. The updates for emission fluxes in Europe by López-Coto et al. (2013) and Karstens



et al. (2015) are not included. The largest terrestrial spatial variation of [222]Rn emission rates in ZKC is a factor of 10.

Figure 1 shows the global annual mean [222]Rn emission fluxes of the four emission scenarios described above. Compared with the standard GEOS-Chem [222]Rn emission scenario (JA97, Fig. 1a), the other three show evident spatial variations of varying extents due to incorporation of observations and estimates from soil exhalation models. The estimated global total [222]Rn emissions for JA97, SW98, ZK11, and ZKC are 1.94 *GCi year$^{-1}$*, 2.41 *GCi year$^{-1}$*, 2.11 *GCi year$^{-1}$*, and 2.22 *GCi year$^{-1}$*, respectively. Since there is no consensus on the global total [222]Rn emission, we do not normalize the total emission amount for each scenario. Instead, the overall evaluation of the emission scenarios is based on comparisons with surface [222]Rn observations. It is clearly shown in Fig. 1 that the three later [222]Rn emission scenarios have substantial enhancements of [222]Rn emission fluxes in North America and East Asia. SW98 exhibits more intense [222]Rn emissions in North America, which have been adopted in ZKC. In the northern polar region, SW98 presents much higher [222]Rn emissions over Siberia extending to higher latitudes. JA97 is overly simplified and has nearly no emissions over Siberia due to temperature-dependent reduction in the cold high-latitude regions. The ZK11 emissions in Siberia stay between those of JA97 and SW98, with somewhat higher emissions in the eastern Siberia. ZK11 has much higher [222]Rn emissions in China, which are further scaled up by a factor of 1.2 in ZKC.

Seasonal variations of [222]Rn emissions are considered in all four scenarios but with different approaches. In JA97, [222]Rn emissions are reduced by a factor of 3 when surface temperature in the driving meteorological fields falls below 0 °C, thus resulting in seasonal variations of [222]Rn emissions in high-latitude regions. In the other emission scenarios, the monthly varying [222]Rn emission fluxes in each model bottom layer are prescribed based on observed and assumed soil parameters (see SW98) and do not change from year to year. Figures 2 and 3 compare monthly mean [222]Rn emissions for January and July in the emission inventories. [222]Rn emissions are generally the lowest in January because of the inhibition of exhalation as a result of ice cover and high moisture content. All emission scenarios exhibit increased

global $^{222}$Rn emissions by a factor of 1.2 to 1.4 in July compared with January due to enhanced emissions over the Northern Hemisphere continents. The summer-winter changes of local emissions are mostly within a factor of 2. The possible underestimation of emissions for surface temperatures under 0 °C is revised in the later emission scenarios, leading to increased wintertime emissions in central and eastern Asia, North

America, and southern Europe (Fig. 2a vs 2d). The affected regions extend to further lower latitudes in eastern Asia and North America compared with relatively warmer Europe.

Compared to JA97, significant emission increases occur in mid-low latitudes where land is covered by desert or mountainous texture in the later emission scenarios, e.g., western U.S. and western China. Rocky and desert land types are more favorable for $^{222}$Rn emission compared with soil. Previous related literature

and analyses support these emission modifications based on evaluations against existing surface observations. We speculate that some degree of emission increase would be reasonable in the Middle East and North Africa where land is mostly covered by desert. Emissions in these areas in ZK11 and ZKC are adapted from SW98, which uses a world average surface radium content to calculate $^{222}$Rn emission. No observations of surface radium content or $^{222}$Rn concentrations exist for evaluating speculated emission

modifications, but future changes are possible depending on the availability of measurements in these areas.

### 2.3 Model simulations and observational data

We simulated $^{222}$Rn with the model driven by MERRA using the four emission scenarios. The preferred emission scenario was then identified based on a comparison of simulated and observed surface

$^{222}$Rn concentrations and seasonality. To characterize convective transport in the MERRA and GEOS-FP products, we also conducted model sensitivity experiments for which convective transport was turned off. All simulations were conducted for the year of 2013 with a 12-month spin-up, which was initialized with a climatological restart file from previous model simulations. Table 1 lists all the model experiments and their configurations.





We evaluate the 2013 simulations against long-term monthly or annual $^{222}$Rn observations. We used the observed surface $^{222}$Rn concentration dataset compiled by Zhang et al. (2011), who evaluated the ZK11 emission scenario in their model. Figure 4 shows the locations of 51 surface $^{222}$Rn measurement sites. The sites are concentrated in Europe, North America, and East Asia. Fewer sites are located in the Southern

Hemisphere. No observations in boreal Canada and Siberia are available. The few in-land sites in China only reported annual means. The $^{222}$Rn observations were made in consecutive years, and we treat the calculated multi-year monthly means as climatological. We also include longer period observations at Mauna Loa (2004-2010; Chambers et al., 2016c) and Gosan (2001-2010; Chambers et al. 2016b) stations in addition to those compiled by Zhang et al. (2011) in our analyses below. Considering the monthly

climatological surface $^{222}$Rn observations used in the comparisons, simulations driven by MERRA for alternative years do not change the conclusions of this study.

To examine simulated convective transport characteristics, we compare model results with four observational datasets of $^{222}$Rn vertical profiles (Liu et al., 1984; Zaucker et al., 1996; Kritz et al., 1998; Williams et al., 2011). Liu et al. (1984) compiled an extensive dataset of $^{222}$Rn profiles for different seasons

based on airborne measurements made in the 1950s-1970s. The summertime average profile was calculated from 23 sites in the U.S., Ukraine, and central Asia, and mainly represents the summertime $^{222}$Rn vertical distribution over the Northern Hemisphere mid-latitude continental regions. Zaucker et al. (1996) reported nine $^{222}$Rn profiles measured during flights from east coast of Canada (Nova Scotia and New Brunswick) to the North Atlantic as part of the North Atlantic Regional Experiment (NARE, August 1993). Kritz et al.

(1998) measured $^{222}$Rn vertical profiles at Moffett Field (37.4 °N, 122.0 °W), a coastal site in California, U.S., during April to August in 1994. The Moffett profile represents summertime $^{222}$Rn vertical distribution over an offshore region. Williams et al. (2011) made aircraft measurements of $^{222}$Rn profiles up to 3.5 km altitude at Goulburn (34.8°S, 149.7°E), an inland rural site in New South Wales, Australia, during May 2006-2008 and January 2007 (Williams et al., 2011).




## 3 Model results and evaluation with surface observations

### 3.1 Model surface $^{222}$Rn

Figure 5a-b show the global surface $^{222}$Rn concentrations for January and July 2013, as simulated by GEOS-Chem with the JA97 emission scenario (simulation A1, Table 1). Surface $^{222}$Rn concentrations are much higher over land at low- and mid-latitudes compared with marine areas and high-latitudes. Typical surface $^{222}$Rn concentrations over land range from a few hundreds to about $1.0 \times 10^4$ *mBq SCM*$^{-1}$. Surface concentrations drop sharply from land to oceanic regions due to the short lifetime of $^{222}$Rn, with the values over the oceans ~2-3 magnitudes lower and ranging from tens to a few hundreds of *mBq SCM*$^{-1}$. The model simulates a noticeable outflow of $^{222}$Rn at surface level from the west coast of Africa to South America in January. Surface $^{222}$Rn concentrations are higher overall in winter due to shallower boundary layers than in summer (Fig. 5a vs Fig. 5b). For example, concentrations in Europe, central and East Asia, and North Africa are significantly lower in July (summer), while concentrations in South Africa, Argentina and Australia are substantially higher in July (winter). The contrasting seasonality of surface $^{222}$Rn concentrations (high in winter) compared to emission fluxes (high in summer) suggests that the accumulation effect in shallower boundary layers (weakened vertical transport and mixing, see Fig. S1) dominates the seasonal changes in emission when it comes to determining the seasonality of surface $^{222}$Rn concentrations.

Figure 5c-h show the changes in simulated surface $^{222}$Rn concentrations when the SW98, ZK11, and ZKC emissions are used (simulation A2-A4, Table 1), relative to the standard simulation with the JA97 emissions. All three alternative $^{222}$Rn emission scenarios lead to remarkable increases in surface concentrations in mid- and high-latitude regions of North America and Asia. With SW98 (simulation A2, Fig. 5c), significantly increased winter surface $^{222}$Rn concentrations appear in northwestern U.S., Alaska, northern Canada, as well as in the continental areas extending from eastern Europe through Siberia to the Bering Strait. These large increases are mainly due to the zero emission flux rate prescribed for high latitudes (> 60°N) in JA97, which is replaced in SW98 by fluxes from 0.3 to 0.6 *atom cm*$^{-2}$ *s*$^{-1}$ (Fig. 2b and

3b). As shown later, this characteristic in JA97 overly simplifies [222]Rn emission variations and causes underestimation of surface [222]Rn concentrations in high-latitude regions in winter. Accumulation of [222]Rn in the shallow winter boundary layer also contributes to and enhances the differences in surface [222]Rn concentration caused by increased emissions. In the ZK11 simulation, similar enhancements of surface

[222]Rn appear in North America, China, and the far East Siberia (Fig. 5e, f), but the overall magnitudes of enhancement are smaller than those with SW98. The largest enhancements in Asia shift to the east and are seen in eastern Siberia rather than the whole boreal Siberia. ZK11 incorporates recent [222]Rn flux measurements in Asia (Zhang et al., 2008) and shows some smaller changes from those of JA97 in Siberia. Since ZKC and SW98 share the same emissions for North America, the surface concentration changes are

mostly identical between the two. For the same reason, the ZKC and ZK11 results look similar in Asia, except that the surface concentrations over China are more enhanced due to the upscaling in ZKC. In July (Fig 5d, f, h), the changes in surface [222]Rn concentrations are much less significant for all emission scenarios. This also reflects the strong effects of summer boundary layer ventilation, which largely compensates for the differences caused by the seasonal emission changes.

**3.2 Evaluation of emission scenarios with surface observations**

Following Zhang et al. (2011), we evaluate the [222]Rn emission scenarios by comparing model results with surface observations of [222]Rn concentrations. We conduct the comparisons in the form of scatter plots for Europe (EU), Asia (AS), North America (NA), and the global (ALL), respectively (Fig. 6). For each observed monthly or annual mean, model output was sampled in the grid cell corresponding to the physical

location and elevation of each site and then averaged for the corresponding observation time period. Also shown in Fig. 6 are the reduced-major-axis linear correlation coefficients (R; Hirsch and Gilroy (1984)) and the percentages (P) of the data points lying within a range of a factor of 2 (dashed lines).

Europe is the continent where emission fluxes and transport of [222]Rn have been studied most extensively. The measurements are more widely and evenly distributed across the continent (Fig. 4). The

JA97 simulation (A1, Table 1) shows moderate agreement with observations (P = 66.5%) bearing some





large underestimates (Fig. 6a). The SW98 simulation has the lowest P value of 61.9% (Fig. 6b) due to a large number of points with high biases. ZK11 and ZKC use the same $^{222}$Rn emissions in Europe, and the P values are close (80.3% and 80.7% in Fig. 6c and Fig. 6d, respectively). The better agreement when using ZK11 and ZKC substantiates the high resolution $^{222}$Rn emission estimates derived from gamma dose rates

in Europe (Szegvary et al., 2009). Schmithüsen et al. (2017) compared measured $^{222}$Rn concentrations across the European sites in terms of different instruments and measurement systems and provided suggested scaling factors for each site. Here, the same evaluation for the emission scenarios with the scaling factors is given in Fig. S2. There are only slight changes in the P values for all regional groups, and the same rank of the four emission scenarios remains.

All simulations exhibit some degree of underestimation in Asia (Fig. 6e-h). Monthly mean observations are available for 7 of the 12 Asian sites, otherwise only annual means are available. Consequently, data points are sparse in Fig. 6e-h. The JA97 simulation shows poor agreement for Asia (P = 46.3%, Fig. 6e). Agreement for the others is better, but still deficient, with P values of 64.2%, 67.2%, and 68.7% for SW98, ZK11, and ZKC, respectively (Fig. 6f-h). The few underestimated data points in Fig. 6g

and 6h are observed annual means from the inland Chinese sites. With upscaled emission in ZKC, the improvement compared with ZK11 is minor. To better match the Asian observations, we tentatively conducted additional model simulations in which the Asian $^{222}$Rn emission fluxes are scaled up by a factor of 1.5 or 1.7 (instead of 1.2 in the recommended ZKC). The P values from those simulations are larger with some previously underestimated data moving into the 2-factor range; up-scaling by a factor of 1.5 would

increase the P value to above 70%, but the simulated total $^{210}$Pb deposition fluxes at mid-latitudes would be substantially overestimated (Zhang et al., Constraints From Airborne Lead-210 Observations on Aerosol Scavenging and Lifetime in a Global Chemical Transport Model, manuscript in prep., 2020). As will be discussed in Section 4, a few studies reported unusually high surface concentrations and excessive emission fluxes at individual sites in Asia; the evidence therein endorsed a higher upscaling factor, which would

reduce the model underestimates of surface concentrations. However, without knowing the distributions



and varying extents of emission biases in Asia, applying a higher and uniform scaling factor to the whole region may worsen the global simulation of $^{210}$Pb deposition. The few annual means that lead to the low P values may not be as representative as the monthly data and can be biased. Therefore, we use a tentative scaling factor of 1.2 for emission fluxes in China (i.e., ZKC) and expect future improvements when more

observations of $^{222}$Rn emission and surface concentrations become available.

All simulations reproduced the observed surface $^{222}$Rn concentrations in North America well (Fig. 6i-l). SW98 (Fig. 6j) and ZKC (Fig. 6l) share identical $^{222}$Rn emissions over North America, and simulations with both emission scenarios show excellent agreement with the observations (P ~ 90%, Fig. 6j, l). This suggests that SW98 is an adequate option for $^{222}$Rn emissions in North America. Interestingly, ZKC leads

to slightly better agreement compared with SW98, although identical emissions were used for North America. A few overestimated data points in the simulation with SW98 are better simulated with ZKC at the U.S. west coast sites, as a result of the large reduction in emissions over the upwind Siberia regions (Fig. 1d). Despite the good agreement between model results and observations, the evaluation is limited to the western and eastern coastal regions of the U.S. Data from Africa, the central U.S., and Canada is

currently lacking, and would otherwise improve the model evaluation, especially over North America.

Figures 6m-p show the overall evaluation of the model results against measurements at all 51 surface sites over the globe. Both ZK11 and ZKC simulations show better agreement with observations (P = 76.9% and 78.4%, Fig. 6o, p), suggesting that ZK11 and ZKC are potentially better choices for replacing the JA97 emission scenario in the standard version of GEOS-Chem. Although with its tentative effort to address high

Asian emissions ZKC is a step ahead of ZK11. The large biases of a few points outside the factor-of-2 range are from the Antarctic sites. None of the simulations can reasonably represent observations in Antarctica, which can be attributed to not well characterized emission (Chambers et al., 2018) and will be discussed later. If the two Antarctic sites (with model low biases in the lower left corner of Fig. 6o, p) were excluded, the P values for ZK11 and ZKC would increase to over 80%.



## 4 Excessive [222]Rn emissions in East Asia

Unusually high [222]Rn emissions have been observed over mainland Asia (Iida et al., 2000; Yamanishi et al., 1991) and downwind regions (e.g., Korea; Zahorowski et al., 2005). These high [222]Rn emissions, not well represented in JA97-like emission scenarios, were likely responsible for the failure of CTMs in capturing the [222]Rn concentrations observed in East Asia (Jacob et al., 1997). In particular, [222]Rn emissions over China have been underestimated at inland cites (Zhang et al., 2011). China and India have been identified as regions of high [222]Rn emissions from soil. It was suspected that this is partially due to high soil content of radium (Schery, 2004; Zhuo et al., 2008). Schery (2004) presented global measurements of radium content in soil, which clearly indicated that the radium concentrations are higher by about a factor of 3 in the southeastern compared to the northwestern China. Consistently, Zahorowski et al. (2005) found that surface [222]Rn concentrations were roughly three times at Hok Tsui (Hong Kong) during winter compared to Gosan, where fetch is from northern China and Mongolia. Zhuo et al. (2008) provided an estimated area-weighted annual average [222]Rn emission of 29.7 $mBq\ m^{-2}\ s^{-1}$ (~1.41 $atom\ m^{-2}\ s^{-1}$) in China. Based on three-year winter-time [222]Rn observations at Sado Island, Japan, and associated trajectory analyses, Williams et al. (2009) suggested that emission fluxes can be 1.75 times higher in the lower latitude bands over the Asian continent compared to higher latitudes. In an inverse modeling of Asian [222]Rn emissions, Hirao et al. (2010) showed an area-weighted average [222]Rn emission of 33.0 $mBq\ m^{-2}\ s^{-1}$ (~1.57 $atom\ m^{-2}\ s^{-1}$) in Asia with the highest emissions found in central and southeastern Asia. These values are considered much higher than typical [222]Rn emission known for Europe, where Szegvary et al. (2009) suggested half of the continent has emissions ranging from 8.33 to 14.6 $mBq\ m^{-2}\ s^{-1}$ (0.40 to 0.70 $atom\ m^{-2}\ s^{-1}$). Hirao et al. (2010) also found that, to better match surface observations at Hachijo Island, a volcanic island about 287 kilometers south of Tokyo in the Philippine Sea, the emissions over East Asia would need to be increased by a factor of 1.69.

It is likely that the high [222]Rn emissions in Asia are poorly estimated because of the diverse climate and geographic textures formed on the largest continent of the earth. The southern part of China is known





to be covered with soils containing higher radium concentrations than global average (Schery and Wasiolek, 1998). Central Asia is dry and sparsely covered with soils, which could facilitate $^{222}$Rn emanation. The mountainous surface in southeastern China could also be conducive to high $^{222}$Rn emissions. The $^{222}$Rn exhalation model developed by Hirao et al. (2010) took into consideration $^{222}$Rn

emission enhancements caused by rough surfaces, but still underestimated $^{222}$Rn concentrations in East Asia. Active crust movements along the east coast of Asia can cause more exposure of radium and extra $^{222}$Rn emissions. Intense human activities may also contribute to excessive $^{222}$Rn emissions in Asia. Moore et al. (1976) pointed out that phosphate ores contain high concentration of $^{238}$U (precursor of radium) and are widely used as phosphate fertilizers in the populous East Asia region. Due to such complexities and

uncertainties, most of the $^{222}$Rn exhalation models are not well validated in Asia, and a lack of $^{222}$Rn measurements in central and western Asia adds to the difficulty.

An alternative way to verify $^{222}$Rn emissions is to evaluate the deposition fluxes of its long-lived decay daughter, $^{210}$Pb. Since surface deposition is the primary sink of $^{210}$Pb aerosols, global $^{210}$Pb deposition fluxes should be balanced by $^{210}$Pb production or $^{222}$Rn emission fluxes (Considine et al., 2005). Regional

total $^{210}$Pb deposition fluxes, however, can be affected by transport into and out of the region. Nevertheless, comparisons between simulated and observed $^{210}$Pb deposition fluxes at multiple locations in Asia offer a test of excessive $^{222}$Rn emissions. Figure 7a compares model results with observed $^{210}$Pb total (dry and wet) deposition in Shanghai for each season averaged over an 8-year period (Du et al., 2015). All model simulations, including the simulation with upscaled emission in China (ZKC), underestimate the total

deposition in all seasons. Enhanced $^{222}$Rn emissions in ZKC improve the simulated $^{210}$Pb deposition to a limited extent in all seasons and more favorably in winter. We then calculate the correlations between simulated and observed annual mean $^{210}$Pb deposition fluxes at the sites in North America (nine sites) and Asia (nine sites; Du et al. (2015)). Details about these sites can be found in Du et al. (2015). The reduced-major-axis regression slopes for North American sites are closer to 1 (Fig. 7b), indicating a generally well

simulated lifecycle from $^{222}$Rn emission to $^{210}$Pb deposition. By contrast, the slopes for Asian sites are





much lower than 1. This large magnitude of model underestimation in $^{210}$Pb deposition fluxes can only be attributed to low $^{222}$Rn emissions in Asia. Much existing evidence suggests using a larger scaling factor, but as mentioned earlier, we choose to use a moderate scaling factor of only 1.2 for China to avoid large overestimates of total $^{210}$Pb deposition fluxes in the Northern Hemisphere.

## 5 Seasonality in surface $^{222}$Rn concentrations

The seasonality in surface $^{222}$Rn concentrations is mainly affected by three factors: (1) the surface $^{222}$Rn emission flux rate determined by radium content and soil conditions; (2) the vertical mixing processes (i.e., boundary-layer mixing and convection); and (3) advective transport of $^{222}$Rn-rich or -poor

air masses. The roles of these factors may vary by location. Here, we examine the seasonal variations of surface $^{222}$Rn concentrations at selected surface sites in Europe, Asia, and North America, and discuss these impacting factors. The selection of surface sites is mainly based on the availability of multiple-year measurements, with consideration of special geographic locations indicative of regional transport patterns.

**Europe**. Observations in Europe were mostly obtained in Finland, Germany, France, and Italy, with

about half of the sites in Finland. Figures 8a-c show the comparisons of model results with monthly mean observations at three Finland sites (Kevo, Pallas, and Joensuu). At these high-latitude sites, the highest monthly concentration does not exceed 4000 *mBq SCM$^{-1}$*, but the seasonal variations are large, with the observed wintertime highs being up to twice the summertime lows. Such seasonal variation is mainly due to shallower boundary layer and less convection in winter because the changes in $^{222}$Rn emissions are

minor due to low temperature all year round (see Fig. 2 and 3). Szegvary et al. (2009) suggested that the $^{222}$Rn emissions in northern Europe are generally lower than the commonly used value of 1 *atom cm$^{-2}$ s$^{-1}$*. The soil water content is high because of the long snowy winter and short summer there. The content of radium is also found to be lower than average in the quaternary sand deposits. The ZK11 and ZKC emission scenarios, which adopted $^{222}$Rn emission fluxes derived from measured gamma radiation

(Szegvary et al., 2009), are clearly the better options and result in better simulated seasonal variations





(frequently overlapped purple and red lines in Fig. 8a-c). The SW98 emissions lead to much higher $^{222}$Rn concentrations compared with the observations, whereas JA97 tends to underestimate the emissions and results in lower concentrations.

Figure 8d-f show model-observation comparisons at three sites in central mainland Europe, i.e.,

Hohenpeissenberg (Germany), Freiburg (Germany), and Gif-sur-Yvette (France). The observations generally show minimal surface concentrations in spring and maximums in late fall. The highs appear earlier with larger seasonal amplitudes compared with the Finland sites as a result of combined effects of seasonal changes in emission fluxes and vertical transport. In general, the lowest $^{222}$Rn concentrations usually occur during spring and summer when convection and boundary layer mixing are most active at

inland surface sites (Wilkening, 1959; Lindeken, 1967). Higher wintertime concentrations at central European sites were also likely attributed to slow transport and long residence time overland due to air mass stagnation (Chambers et al., 2016a; Williams et al., 2016). At mid-latitude sites, snow cover suppresses $^{222}$Rn exhalation and reduces emission fluxes substantially in winter; complete snow melt and moist flux enhance $^{222}$Rn emissions in summer (Reithmeier and Sausen, 2002). Since strong emissions in

summer partially compensate the dilution effect of boundary-layer mixing and strong convection, the lowest $^{222}$Rn concentrations are usually observed in the springtime. All simulations capture the seasonal variations; ZK11 and ZKC emission scenarios do not lead to obviously better results than JA97 and SW98. It appears that a sharp increase in emission is missing from summer to late fall as indicated by increased observations in June-August, suggesting that further emission adjustments are needed for Europe in the

model. Szegvary et al. (2009) also suggested large $^{222}$Rn emissions over the Iberian Peninsula and the northern Mediterranean coastal region due to a wide coverage by dry soil and crystalline rocks. In a more recent study using $^{222}$Rn as a tracer to classify atmospheric stability in Slovenia, unusually large $^{222}$Rn exhalation flux from flysch and carbonate rocks at an inland site was found to cause higher $^{222}$Rn concentrations in the diurnal cycle compared to a costal site where atmospheric synoptic conditions were

considered similar but land was more dominated by sea and lake sediments (Kikaj et al., 2019).



Figure 8g shows the model-observation comparison for Mace Head, a coastal site in Western Europe (Ireland). Most observations are lower than 1000 *mBq SCM$^{-1}$* with a weak seasonal variation. Simulations with JA97 and SW98 overestimate the observations with fluctuated seasonality. The coastal site is usually moist and largely affected by oceanic air; it is therefore characterized by relatively low $^{222}$Rn

concentrations all year round. A regional model simulation by Chevillard et al. (2002) with a JA97-like, uniform $^{222}$Rn emission rate, showed similar overestimation with much larger discrepancies from observations. The site is located (53.3°N) very close to the cut-off latitude (60°N) in JA97, at which zero emissions are assumed northwards. The comparisons in Europe suggest that the fixed emission fluxes (with reductions under freezing conditions) in JA97 can lead to overestimation in southern Europe,

underestimation in the north, and a weaker latitudinal gradient towards the north as shown by ZK11 and ZKC is much favored. The comparisons with measurements applied with scaling factors suggested by Schmithüsen et al. (2017) are given in Fig. S3, which only shows slight changes.

**Asia**. Observations of surface $^{222}$Rn concentrations in Asia, e.g., southern China (Zahorowski et al., 2005), Japan (Chambers et al., 2009; Iida et al., 2000), and India (Debaje et al., 1996), are clearly affected

by Asian summer monsoon, with maximum concentrations observed in winter and minimums in summer (low-$^{222}$Rn marine air brought by the monsoon). Figure 9 shows the model-observation comparisons at five Asian sites (Beijing, Gosan, Fuzhou, Hong Kong, and Bombay). Inland sites in China, where only annual mean observations are available, are not included in this comparison. The observations at Beijing show a moderate seasonal variation similar to the mid-latitude continental European sites, with a spring minimum

and an autumn maximum. The simulation with JA97 shows reasonable agreement with observations at Beijing only in spring and summer, but is significantly biased low in late fall-early spring (November-March, Fig. 9a). The latter is likely due to the temperature-dependent reduction of $^{222}$Rn emissions in JA97 when surface temperature is below 0°C. In reality, soil may not be frozen when temperature remains below 0°C for a short period of time. At Gosan, an island site largely affected by Asian monsoon and emissions

from the major Asian continent, observations show a strong seasonal variation with a winter maximum and





a summer minimum. The large winter low bias at Gosan with JA97 is likely also due to the assumed dependency on surface temperature.

At two southeastern China sites, Fuzhou and Hong Kong, the model largely underestimates the observations (Fig. 9c and 9d). The $^{222}$Rn observations show a minimum in summer, reflecting the intrusion of low-$^{222}$Rn marine air associated with the Asian summer monsoon. Although the model successfully captures the seasonal variation, all simulations underestimate observed values all year round, especially at Fuzhou (Fig. 9c). The simulation with ZKC (with enhanced emissions in China) also results in large underestimation. This is likely attributed to the unusually high emission fluxes in southeastern China possibly due to the rocky texture in the mountainous region. On the other hand, the simulations with ZK11 and ZKC capture the observations at Bombay, India, well. These contrasting model performances suggest that $^{222}$Rn emission fluxes in southeastern China need to be better quantified with flux measurements at more surface sites.

**North America**. Figure 10 shows the model-observation comparisons at four U.S. continental sites. Similar to those mid-latitude surface sites in Europe and Asia, the observations at the U.S. sites show seasonal lows in spring and highs in fall or winter. The simulations with SW98 and ZKC (identical emissions over North America) show much higher $^{222}$Rn concentrations than those with JA97 and ZK11 over the U.S. The seasonality at Chester is well captured by using SW98 and ZKC. At Cincinnati, the model performs slightly better with JA97 and ZK11, while the simulations with SW98 and ZKC overestimate the autumn peaks by nearly a factor of two. SW98 and ZKC lead to significant positive biases at Washington D.C., even though ZK11 commits negative biases of a similar magnitude. At Socorro, an elevated site (1400 m a.s.l.) in southern U.S., all simulations hardly capture the seasonal variation (Fig. 10d). Socorro is located in the Rio Grande Valley, where $^{222}$Rn emissions may have larger variations due to surface textures and local meteorology (e.g., upslope air flows) that cannot be resolved by the coarse resolution model.





**Other Sites**. Figure 11 shows the seasonal variations of surface $^{222}$Rn concentrations at eight sites in remote areas or the Southern Hemisphere. Surface $^{222}$Rn concentrations at Bermuda show a late spring to summer minimum (May-August) due to the strengthened Azores-Bermuda High pressure system in summer which brings low-$^{222}$Rn air from the central and eastern North Atlantic (Fig. 11a). At Mauna Loa,

observations are in a low range of 75-150 *mBq SCM$^{-1}$* all year round, reflecting low $^{222}$Rn in marine free tropospheric air (Fig. 11b). The seasonality is, however, distinct with minimum in summer and maximum in late-winter/early-spring when efficient monsoonal transport of continental air occurs (Balkanski et al., 1992; Zahorowski et al., 2005). At both remote sites, the model captures the seasonality reasonably. The seasonal amplitudes in all simulations are larger than observed, except with JA97. The simulation with

JA97 better captures the observed amplitude but substantially underestimates the concentrations. It is challenging for a coarse-resolution global model (with unresolved topography and grid-averaged local emissions) to accurately simulate the low $^{222}$Rn concentrations at such a remote island.

Figures 11c-f show the $^{222}$Rn seasonality at three subtropical sites, Chacaltaya (Bolivia), Rio de Janeiro (Brazil), Cape Point (South Africa; Botha et al., 2018), and one mid-latitude site, Cape Grim (Australia) in

the Southern Hemisphere. Seasonal variations are similar to the Northern Hemispheric sites, showing highs in winter and lows in summer. The model fails to reproduce the observed seasonal trend at Chacaltaya, presumably due to its high elevation (5421 m a.s.l. on the Andes) that is not well resolved. At the two Antarctic sites (Fig. 11g, h), the model does not simulate well the seasonal variations likely due to a lack of emission measurement and oversimplified emission fluxes. With all emission scenarios except SW98, the

model underestimates the observations substantially during warmer seasons (November to February), as also noted by Zhang et al. (2011). In fact, snow (ice) melting and reforming may enhance $^{222}$Rn emissions and surface concentrations in relatively warmer seasons. SW98 is the only scenario with prescribed non-zero emission fluxes in the Antarctic. It arbitrarily assigns a small fixed value to the emission in the Antarctic region due to no measurements of soil $^{226}$Ra content, but causes model overestimates in surface





$^{222}$Rn concentrations at the two sites, especially during winter. More future measurements of $^{222}$Rn emissions in Antarctic regions are thus desired.

## 6 Vertical distribution of $^{222}$Rn concentrations

The vertical distribution of $^{222}$Rn reflects mainly the convective transport process rather than large-scale advection due to the relatively short decay lifetime (a few days) of $^{222}$Rn. However, it is more challenging for global models to capture the convective transport of $^{222}$Rn concentrations to the middle and upper troposphere than the synoptic-scale transport (Jacob et al., 1997). In this section, we characterize the convective transport in GEOS-Chem driven by the MERRA and GEOS-FP meteorological data sets,

respectively, and evaluate model simulations with observed $^{222}$Rn vertical profile.

### 6.1 Simulated $^{222}$Rn profiles and comparison with observations

The most widely used $^{222}$Rn profile measurements were compiled by Liu et al. (1984) (black line, Fig. 12a). The composed profile is averaged from $^{222}$Rn observations over the U.S., Ukraine, and central Asia and represents the summer $^{222}$Rn vertical distributions over northern mid-latitude continental regions. The

profile shows an inflection point between 3 and 4 km, reflecting the average altitude of convective entrainment (Fig. 12a). Concentrations decrease slowly as altitude increases from 4 to 7 km, indicating fast convective transport over land during summer (Liu et al., 1984; Zhang et al., 2008). We sample the simulated monthly mean $^{222}$Rn profiles at the provinces or states where each observed profile was measured, and obtain an average profile for each simulation. As shown in Fig. 12, all simulations well

capture the rapid decrease of $^{222}$Rn concentrations from the surface to about 4 km at a rate of 1000 *mBq* *SCM$^{-1}$* per *km*. The simulated concentrations then decrease faster than the observations until 6 km. It is suggested to be a consequence of overly vigorous convective transport in the model (Considine et al., 2005). MERRA exhibits a higher and deeper convection from 5 to 10 km. As a result, a remarkable underestimation of $^{222}$Rn concentrations with MERRA is seen from 4 to 8 km, followed by overestimations

above 9 km. Due to weaker convection in GEOS-FP, the simulation underestimates in a broader altitude (4-



10 km). It seems challenging for the GEOS products to capture the convective detrainment level. As pointed out below, weaker convection in GEOS-FP at the resolution of 2° × 2.5° is partially due to the transport errors introduced by using the archived and regridded meteorological data (Yu et al., 2018).

Figure 12b compares model results with the $^{222}$Rn profile averaged from measurements obtained at Moffett Field, a coastal site in California, U.S., during June to August in 1994 (Kritz et al., 1998). The model profiles are obtained by averaging monthly $^{222}$Rn concentrations in the grid column corresponding to the site and those in the grid column to the west as suggested by Zhang et al. (2008). The simulations hardly capture the "C" shape profile, a sign of strong convective transport in summer. The simulation with JA97 performs better until up to 5 km, above which those with ZKC and SW98 agree better with the observations. The large overestimation at 2 to 5 km with ZKC and SW98 is likely due to too strong shallow convection and/or high emission fluxes. The differences in near-surface concentrations between the simulations with ZKC and SW98 (Fig. 12b) are caused by averaging ZK11 and SW98 emission fluxes along the edges of the continent in the formulation of ZKC.

Figure 12c shows the comparison of model simulations with the profile averaged from aircraft measurements in the east coastal region of Canada during NARE in August 1993 (Zaucker et al., 1996). The model results are averages over a region of 41-46°N and 60-70°W. The simulation with JA97 reasonably reproduces the observations between 0-4 km, while the simulation with ZKC overestimates between 2 and 5 km. The model performance for NARE is similar to that for Moffett Field. The stronger emissions (ZKC and SW98) tend to result in overestimates in the lower free troposphere (Moffett Field and NARE) but better estimates in the upper troposphere (Moffett Field).

The vertical $^{222}$Rn profiles at Goulburn were measured up to about 3.5 km above the ground level in May of 2006-2008 (Fig. 12d; Williams et al., 2011). The corresponding model results are monthly means for May of the simulation year. The model underestimates the concentrations substantially but well simulates the vertical gradient. It suggests that the underestimation is more likely caused by potentially low biases in the emissions over the Australian continent rather than errors associated with vertical mixing in





the model. Despite this, the model reproduces well the seasonality in surface $^{222}$Rn observations at Cape Grim (Fig. 11f), which is located on the Tasmania Island to the south of the Australian continent. Above 2.5km, the vertical gradient of $^{222}$Rn concentrations decrease in both the observations and the model.

Two model uncertainties may affect our simulated $^{222}$Rn profiles: the remapping of the meteorological data from the original cubed-sphere grid in the parent GCM (GEOS-5) to an equally rectilinear (latitude-longitude) grid in the off-line CTM (GEOS-Chem); the degradation of the temporal and spatial resolutions of the meteorological archive (Yu et al., 2018). Yu et al. demonstrated that such remapping and using 3-hourly averaged wind archives may introduce 5-20% low biases into vertical transport of $^{222}$Rn, including the weakened transport from the boundary layer to the upper troposphere. They also showed that degrading the spatial resolution of the meteorological archive for input to GEOS-Chem further weakened vertical transport because organized vertical motions in the finer resolution are averaged out in the coarser resolution. Such biases may partially contribute to the discrepancies between the simulated and observed $^{222}$Rn profiles, which appear to be larger in the mid- and upper- troposphere (5.5-10 km) when the model is driven by GEOS-FP (Fig. 12). GEOS-FP has finer native horizontal resolution (0.25° latitude by 0.3125° longitude) than MERRA reanalysis (0.5° latitude by 0.667° longitude). An effort is currently ongoing to restore the lost vertical transport by implementing the modified Relaxed Arakawa-Schubert convection scheme in GEOS-Chem (He et al., 2019).

**6.2 Role of convective transport: MERRA vs. GEOS-FP**

To examine the role of convective transport in simulated distributions of $^{222}$Rn, we compare model simulations driven by MERRA and GEOS-FP where the convective transport operator is turned on or off (i.e., A1, B1, A1-nc, and B1-nc, where "nc" denotes no convection; Table 1). Figure 13 shows the latitude-pressure cross-section of annual zonal mean $^{222}$Rn concentrations in these four simulations. The concentrations are contoured on a logarithmic scale. The strong gradients above the tropopause in all panels are indicative of a fast decrease of $^{222}$Rn concentrations due to weak vertical diffusion. The interhemispheric asymmetry in $^{222}$Rn distributions reflect the larger landmass and $^{222}$Rn emissions in the





Northern Hemisphere. The latitudinal and vertical distributions of $^{222}$Rn concentrations simulated with

MERRA and GEOS-FP are very similar. The overall vertical transport in the simulation with MERRA is

slightly stronger than with GEOS-FP as shown by the higher $^{222}$Rn concentrations near the tropical

tropopause (Fig. 13a-b). In contrast, when convection is turned off (Fig. 13c-d), the model simulates higher

$^{222}$Rn concentrations near the tropical tropopause with GEOS-FP than with MERRA, indicating that

convection is stronger in MERRA than in GEOS-FP.

Figure 14 shows the relative contributions to $^{222}$Rn annual zonal mean concentrations by convection in

MERRA and GEOS-FP, defined as $(^{222}Rn - ^{222}Rn_{nc})/^{222}Rn \times 100\%$ where $^{222}Rn$ and $^{222}Rn_{nc}$ denote

simulations with and without the convection operator, respectively. Where positive values occur,

convection facilitates the transport of $^{222}$Rn to the region and increases $^{222}$Rn concentrations. Similarly,

negative values indicate convection decreasing $^{222}$Rn concentrations. The negative values in the lower

troposphere along with the positives in the middle and upper troposphere are due to the pumping effect of

convection, transport surface-emitted $^{222}$Rn upward. Convection in the simulation with GEOS-FP transports

about 20-30% less $^{222}$Rn to higher altitudes in the tropics and subtropics compared to MERRA (Fig. 14a vs

14b). Figure 15 shows the annual mean convective and large-scale vertical fluxes of $^{222}$Rn in the

simulations with MERRA and GEOS-FP as well as their differences. Convective fluxes are stronger in a

broader latitude range (30°S-55°N) in the simulation with MERRA. The largest difference appears in the

tropical lower troposphere where convective fluxes of $^{222}$Rn in the simulation with MERRA are about a

factor of two larger than those in the simulation with GEOS-FP (Fig. 15c). The large-scale vertical fluxes

of $^{222}$Rn in the simulation with GEOS-FP are significantly larger than those with MERRA (Fig. 15f), partly

compensating the differences in convective fluxes. This compensation leads to the aforementioned general

similarity in the zonal mean $^{222}$Rn distributions in the two simulations (Fig. 13).

To further illustrate the differences in convective transport between the simulations with MERRA and

GEOS-FP, we show in Fig. 16 the simulated $^{222}$Rn profiles averaged over the northern mid-latitude land

areas (30°-60°N) for both cases of with and without convection. The solid black line with the upper x-axis



presents the corresponding concentration ratios between the two simulations. Similar to the earlier analysis of $^{222}$Rn vertical fluxes, convection in MERRA is stronger as indicated by the large change in $^{222}$Rn concentrations at high altitudes (e.g., 8 km) when convection is off (solid red line vs. dashed red line). The different characteristics of vertical transport in MERRA and GEOS-FP are better revealed by examining

the $^{222}$Rn concentration ratio profiles (black and green lines with the upper x-axis). Convective transport takes effect from the base of cloud layers (i.e., the lowest model layer with non-zero convective mass fluxes) in the model, whereas the large-scale vertical advection occurs from the bottom model layer up. As shown by $^{222}$Rn concentration ratios between the two simulations with convection turned off (green line, Fig. 16), it is more efficient in GEOS-FP than in MERRA to transport $^{222}$Rn vertically through large-scale

advection and boundary-layer mixing ($^{222}$Rn ratios < 1 above ~2.5 km and > 1 below). Even with convection turned on, simulated $^{222}$Rn concentrations near the surface are still lower in GEOS-FP than in MERRA (solid black line, Fig. 16) because large-scale advection and boundary-layer mixing dominate near surface and drain surface $^{222}$Rn faster. When $^{222}$Rn reaches the base of convective clouds, it is more efficiently uplifted in MERRA due to stronger convection, resulting in lower $^{222}$Rn concentrations in the

lower troposphere ($^{222}$Rn ratios < 1 from ~0.75 to 4 km), and higher concentrations in the middle to upper troposphere (> 4 km). This feature should also affect the simulations of other surface emitted species when using MERRA and GEOS-FP as the driving meteorology in GEOS-Chem.

## 7 Summary and conclusions

We have evaluated the global distributions of $^{222}$Rn simulated by the GEOS-Chem chemical transport model with a focus on the sensitivity of simulated surface concentrations and seasonality to the choice of available emission scenarios. A preferred emission scenario was recommended based on evaluations against surface observation of $^{222}$Rn concentrations and $^{210}$Pb deposition fluxes. We have discussed the major factors controlling $^{222}$Rn emissions as well as potential emission uncertainties in East Asia, North

Africa, and Antarctic. We have also characterized the vertical transport processes associated with the


MERRA and GEOS-FP meteorological data products by comparing simulated [222]Rn vertical profiles with observations.

We implemented three new global [222]Rn emission scenarios in GEOS-Chem, SW98 (Schery and Wasiolek, 1998), ZK11 (Zhang et al., 2011), and ZKC (an optimized inventory modified from Zhang et al. (2011)). All scenarios include prescribed regional variations and seasonality, which are lacking in the JA97 emission scenario (Jacob et al., 1997) currently used in the standard GEOS-Chem and other global models. JA97 often led to much larger biases in surface concentrations relative to the other scenarios because of lack of spatial variations and overly simplified emission reduction under freezing conditions (e.g., in high-latitude regions). The new emission options all resulted in remarkable increases in surface [222]Rn concentrations at northern mid- and high-latitudes. Such increases were more pronounced in winter due to the accumulation effect within the shallow boundary-layer. With constraints from observations, we are able to achieve much better agreements between the model and observations in all four defined regions (Europe, Asia, North America, and remote regions) using a customized emission scenario, ZKC. However, the simulation with ZKC still inherited some unsolved issues, e.g., large biases in Asia, poorly characterized emission fluxes in Antarctica, and at some elevated sites. More measurements of soil radium content and surface [222]Rn concentrations are desired to produce a better global [222]Rn emission scenario. The seasonality in surface [222]Rn concentrations at northern mid-latitudes typically shows a low in spring and a peak in fall, a result of the competition between changes in emission fluxes and the strength of vertical transport (ventilation). In subtropical East and South Asia, the seasonality is strongly affected by monsoon and shows a summer minimum. Our analyses also suggested that [222]Rn emissions have been quantified more accurately over Europe due to more frequent and evenly distributed measurements across the continent.

We specifically investigated the excessive Asian [222]Rn emissions and explored possible reasons based on previous studies. Both our simulated surface [222]Rn concentrations and [210]Pb (decay daughter of [222]Rn) deposition fluxes over Asia suggested excessive [222]Rn emissions in Asia. In the simulation experiments with Asian [222]Rn emissions scaled up by a factor of 1.2 to 1.7, agreements with surface observations were



significantly improved. However, due to limited knowledge about the spatial distributions and extents associated with the underestimation in Asian emissions, we did not apply a larger scaling factor, which would cause large overestimates of $^{210}$Pb deposition fluxes in the model. As a trade-off, we used a scaling factor of 1.2 for emissions over China in the ZKC inventory, which increased the simulated surface $^{222}$Rn

concentrations and led to a better agreement with observations in Asia. The issue of underestimated Asian emissions is still open. An ideal solution would be an improved and spatially resolved emission map instead of using a uniform scaling factor for the region. The excessive $^{222}$Rn emissions in Asia may be due to multiple factors, including various surface textures, high contents of radium in the soil, active crust movement along the Asian earthquake zone, and high contents of radium in the fertilizer used in East Asia

and India.

We found that it was challenging for model simulations driven by GEOS products to fully capture the vertical structure of observed $^{222}$Rn profiles. A comparison with summertime continental profiles showed that both MERRA and GEOS-FP have biased levels of convective detrainment. Convection in both MERRA and GEOS-FP was likely too deep in northern mid-latitude land areas. The strength of convection

in GEOS-FP is too weak, leading to large low biases of $^{222}$Rn in the mid-high troposphere. This is partly attributed to the lost vertical transport as a result of the remapping from the cubed-sphere to equally rectilinear grids and the degradation of the spatiotemporal resolution of the input meteorological data (Yu et al., 2018). A comparison of global $^{222}$Rn vertical distributions between the simulations driven by MERRA and GEOS-FP showed a distinct difference in the role of convective transport (versus large-scale

vertical advection) in determining the $^{222}$Rn vertical distributions. The stronger convective transport in MERRA is partially compensated by its weaker large-scale upward advection compared with GEOS-FP, resulting in similar vertical $^{222}$Rn distributions in the model simulations driven by the two meteorological products. This has important implications for using chemical transport models to interpret the transport of other trace gases and aerosols when these GEOS products are used as driving meteorology.



*Data availability.* The $^{222}$Rn emission data used in this paper is described in Section 2.2. Observational data for model evaluation are introduced in Section 2.3. All model output, emission data, and observational data sets are available online (http://doi.org/10.5281/zenodo.3942287).

*Author contribution.* BZ and HL designed the study. BZ conducted the model simulations and led the analysis. BZ and HL wrote the manuscript with contributions from all coauthors. SC, CHK, and AGW contributed $^{222}$Rn surface and profile data sets. KZ contributed the ZK11 $^{222}$Rn emission data and surface observational data sets. MPS and RMY contributed to the model development.

*Competing interests.* The authors declare that they have no conflict of interest.

*Acknowledgements.* This work was funded by the NASA Atmospheric Composition Campaign Data Analysis and Modeling program (ACCDAM, grant number NNX14AR07G) managed by Hal Maring. HL would like to thank Daniel Jacob for his comments on an earlier version of the manuscript, and Andrea Molod for useful discussions. The Pacific Northwest National Laboratory is operated for DOE by Battelle Memorial Institute under contract DE-AC06-76RLO1830. NASA Center for Computational Sciences (NCCS) provided supercomputing resources. The GEOS-Chem model is managed by the Atmospheric Chemistry Modeling Group at Harvard University with support from NASA ACMAP and MAP programs. The GEOS-Chem Support Team at Harvard University and Dalhousie University are acknowledged for their effort.

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





**Tables and figures**

**Table 1. Configurations of GEOS-Chem simulations (v11-01f, 2°×2.5°) used in this work.**

| Simulation | $^{222}$Rn emission | Driving Meteorology | Convection |
|---|---|---|---|
| A1 | JA97 (Jacob et al., 1997) | MERRA | on |
| A2 | SW98 (Schery and Wasiolek, 1998) | MERRA | on |
| A3 | ZK11 (Zhang et al., 2011) | MERRA | on |
| A4 | ZKC (this work) | MERRA | on |
| B1 | JA97 (Jacob et al., 1997) | GEOS-FP | on |
| A1-nc | JA97 (Jacob et al., 1997) | MERRA | off |
| B1-nc | JA97 (Jacob et al., 1997) | GEOS-FP | off |

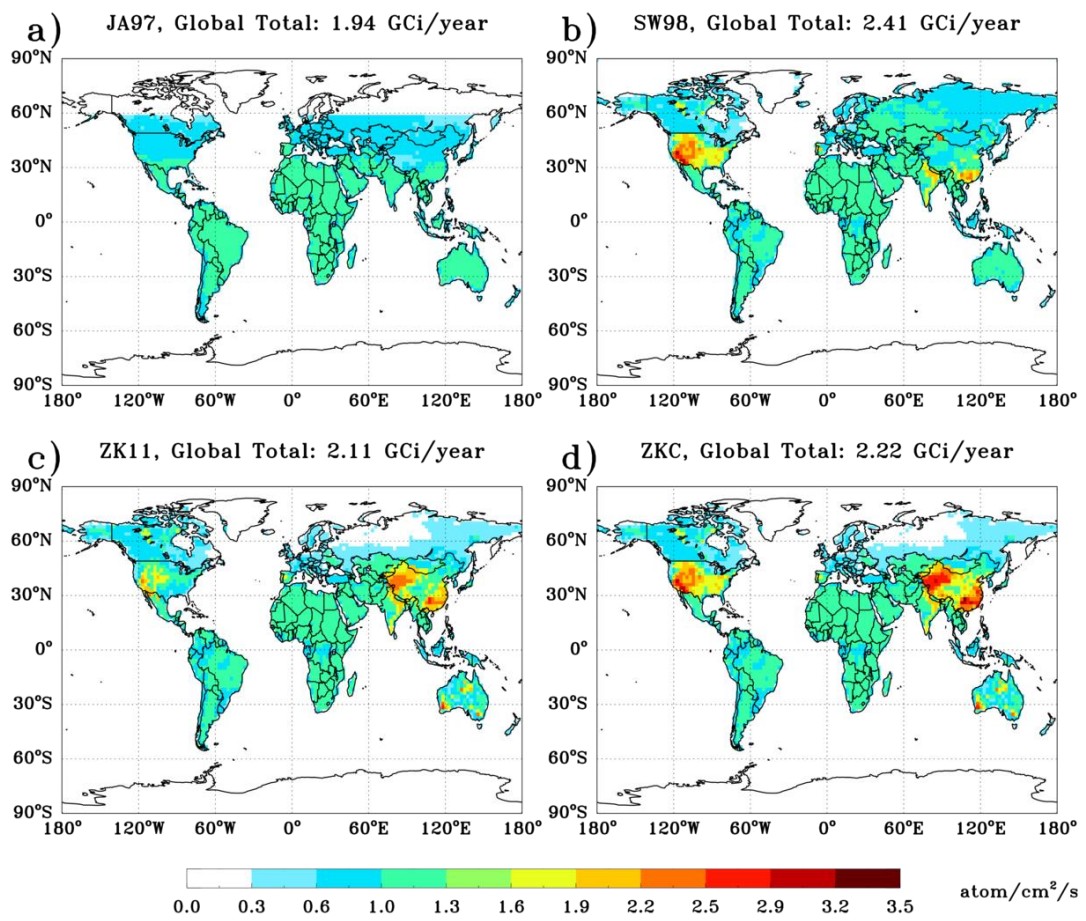

Figure 1. Global annual mean surface $^{222}$Rn emission fluxes (atom cm$^{-2}$ s$^{-1}$) of four emission scenarios used in this study. a) JA97: the default emission in the standard version of GEOS-Chem (Jacob et al., 1997); b) SW98: the first global $^{222}$Rn emission with regional variability based on a soil emission model (Schery and Wasiolek, 1998); c) ZK11: a recently published global $^{222}$Rn emission combining SW98 and recent
10 measurements of $^{222}$Rn fluxes (Zhang et al., 2011); and d) ZKC: ZK11 with customized adjustments to better match observations (this work).





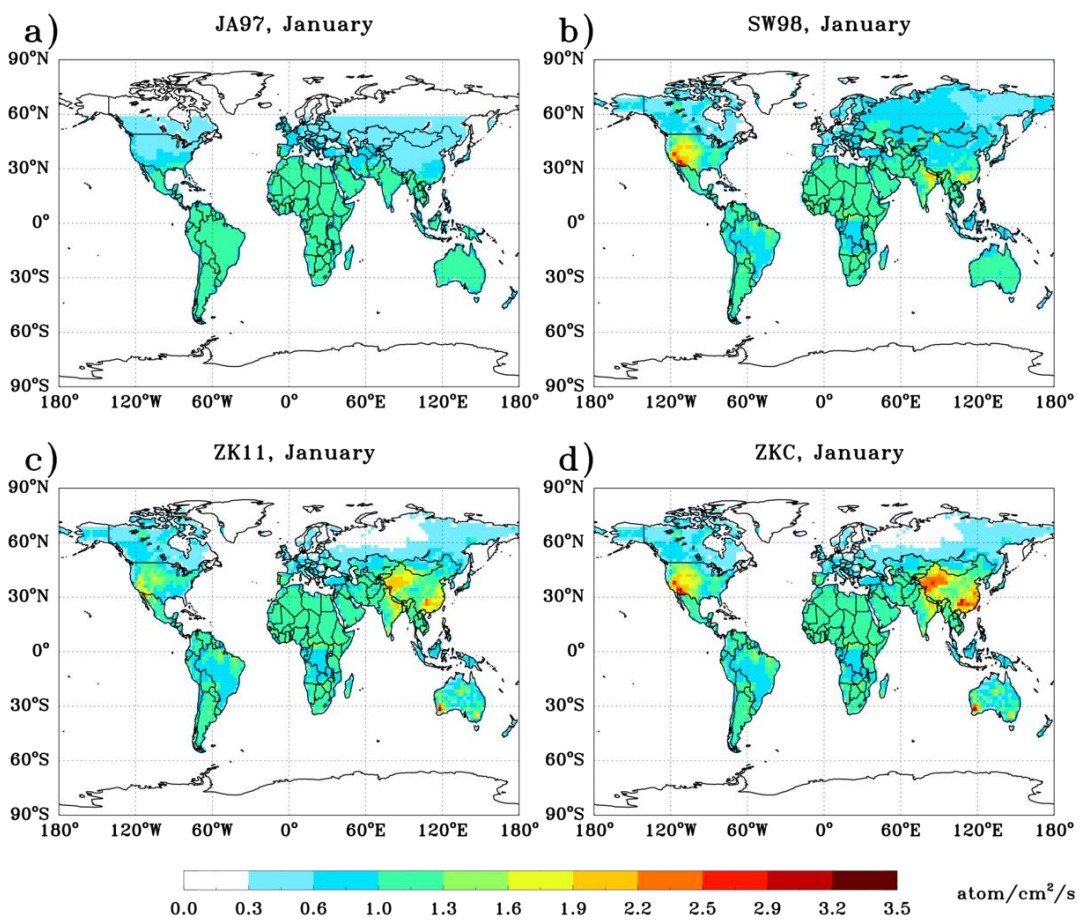

Figure 2. Same as Figure 1 but for January.





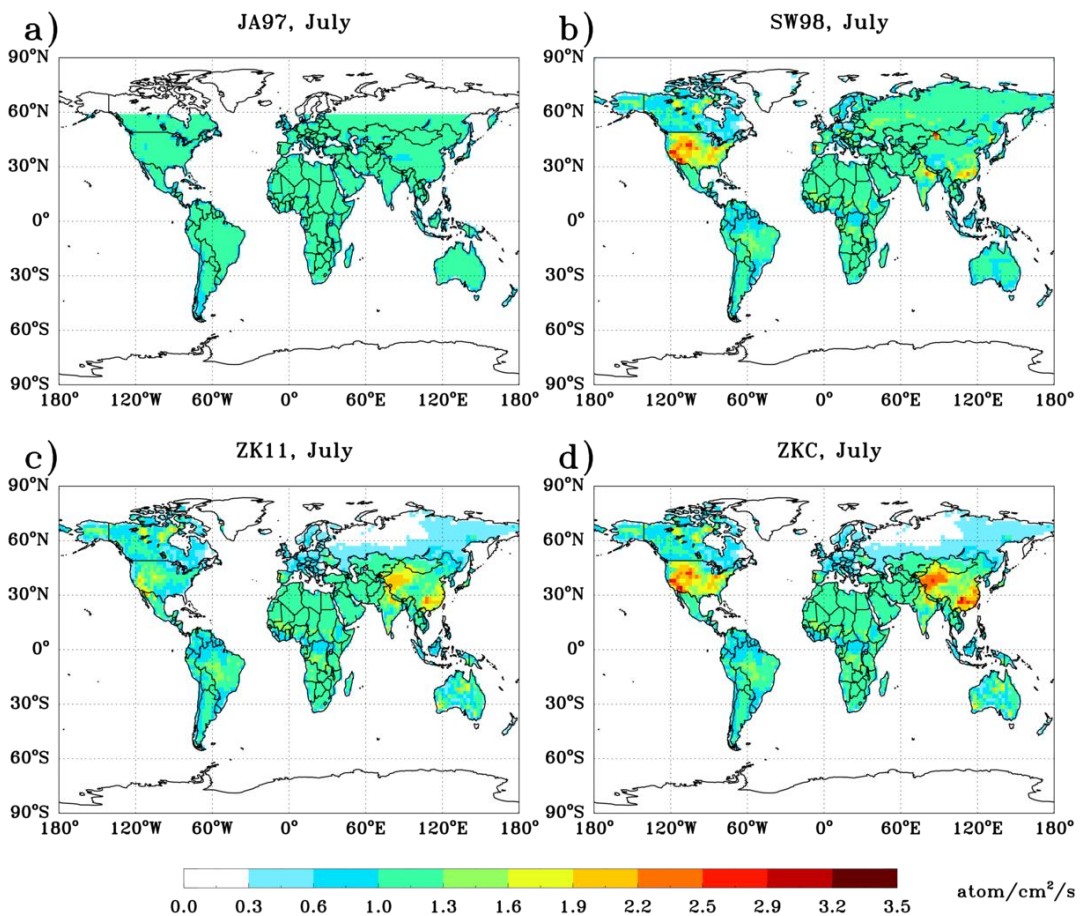

Figure 3. Same as Figure 1 but for July.





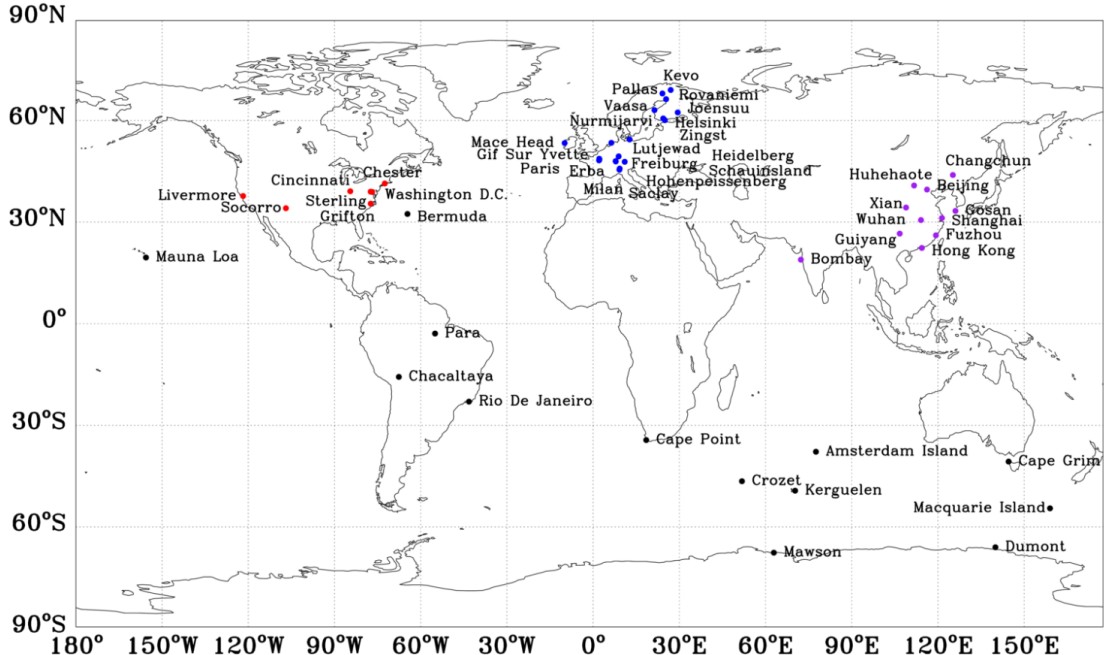

Figure 4. Locations of surface $^{222}$Rn measurement sites. Sites in four distinctive regions are color-coded: Europe (blue), Asia (purple), North America (red), and remote regions (black). Refer to Table 2 of Zhang et al. (2011) for more details.



Figure 5. Simulated monthly surface $^{222}$Rn concentrations (mBq/SCM) for (a) January 2013 and (b) July 2013 with the JA97 emission scenario (A1, see Table 1). Panels (c)-(h) are same as (a) and (b) but showing the changes in surface $^{222}$Rn concentrations when SW98 (A2), ZK11 (A3), and ZKC (A4) emissions are used in the model, respectively.

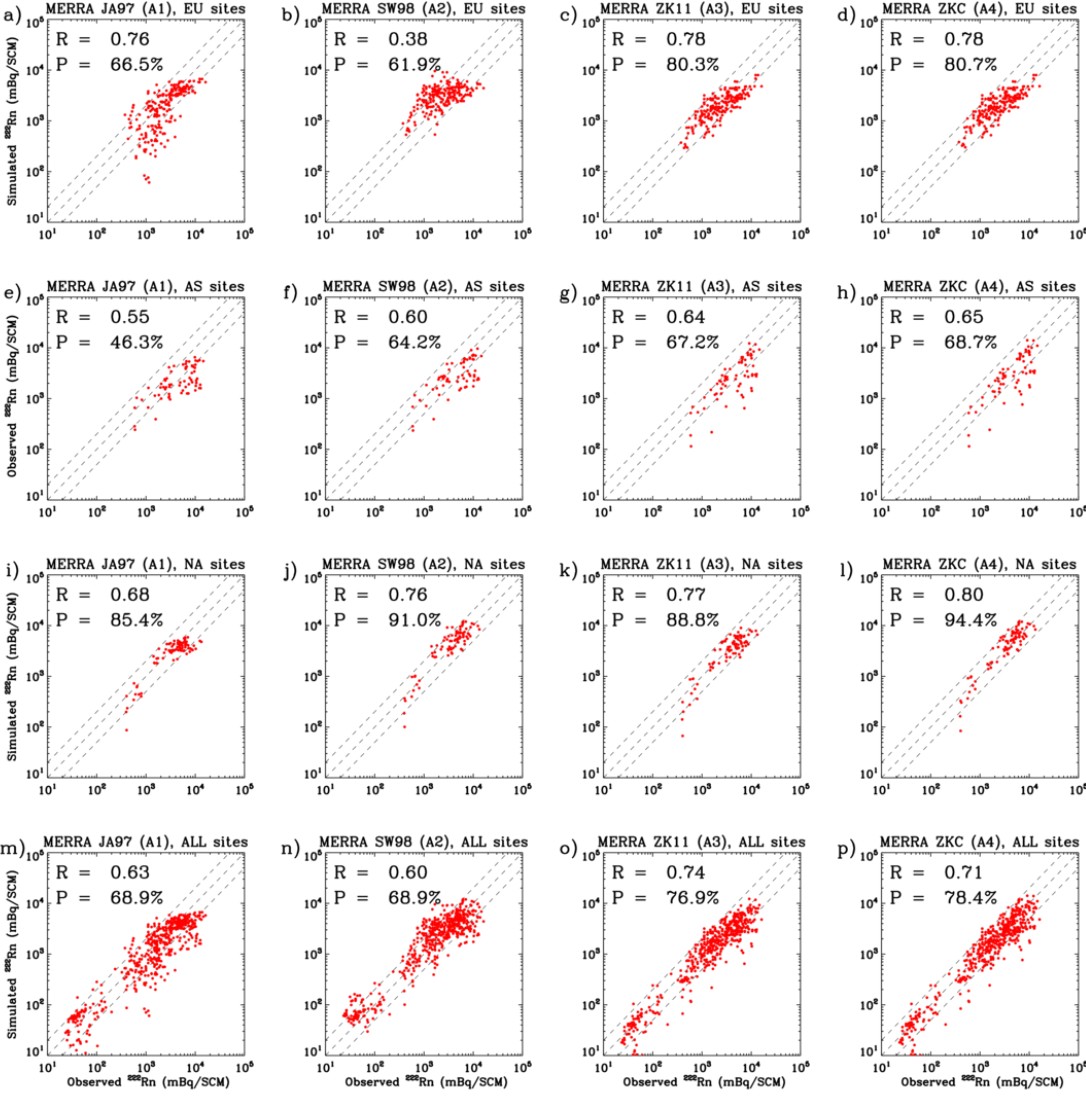

Figure 6. Comparisons between simulated and observed monthly surface $^{222}$Rn concentrations (mBq/SCM)
over the continents of Europe (EU, first row), Asia (AS, second row), North America (NA, third row), and
over the globe (ALL, last row), respectively. The four columns correspond to simulations (A1-A4) with the
four emission inventories (JA97, SW98, ZK11, and ZKC, see Table 1). Dashed lines indicate the range
with a factor of two of the 1 to 1 line. P is the percentage of samples within this range.  R in the legends is
the 2-sided linear regression correlation coefficient. The lines of best fit are calculated using the reduced-
major-axis method (Hirsch and Gilroy, 1984).



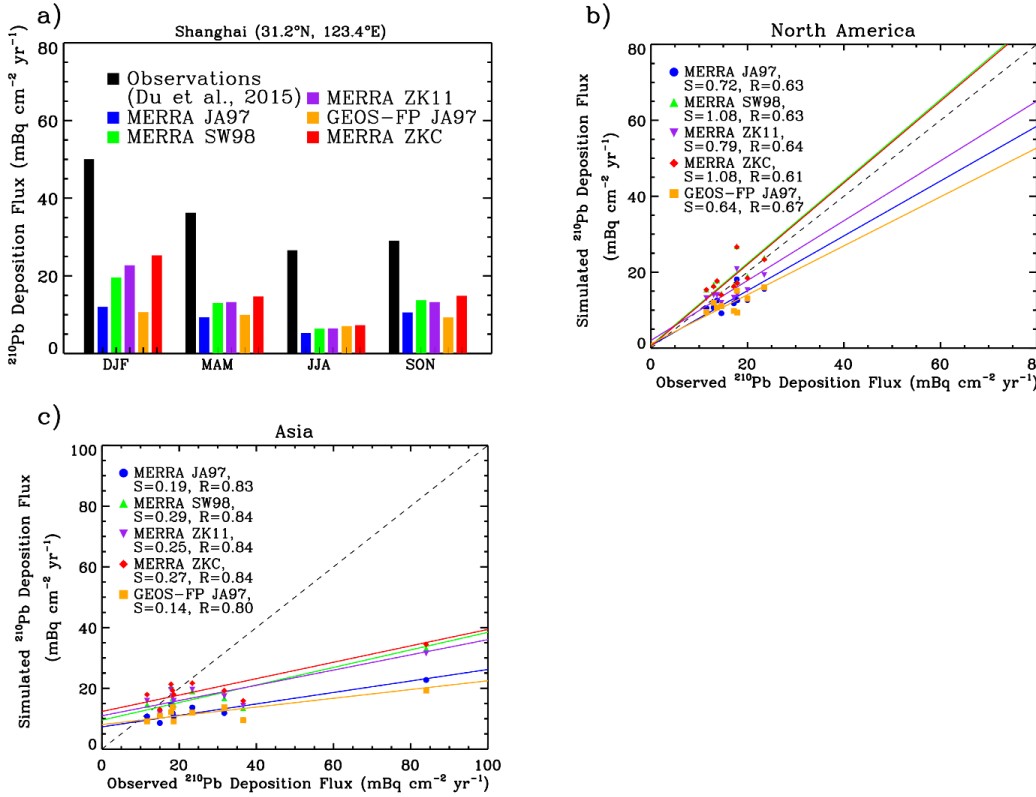

Figure 7. a) Comparison of seasonal total $^{210}$Pb deposition fluxes (mBq cm$^{-2}$ yr$^{-1}$) at Shanghai (32.1°N, 123.4°E) between five model simulations (see Table 1) and observations (Du et al., 2015). b) Correlations between simulated and observed $^{210}$Pb deposition fluxes at nine surface sites in North America (Du et al., 2015). c) Same as b) but for nine Asian sites. Dashed line is the 1 to 1 line. Color lines are linear regression lines for the five model simulations shown in the legends. The reduced-major-axis regression slopes (S) and correlation coefficients (R) are given in the legends.

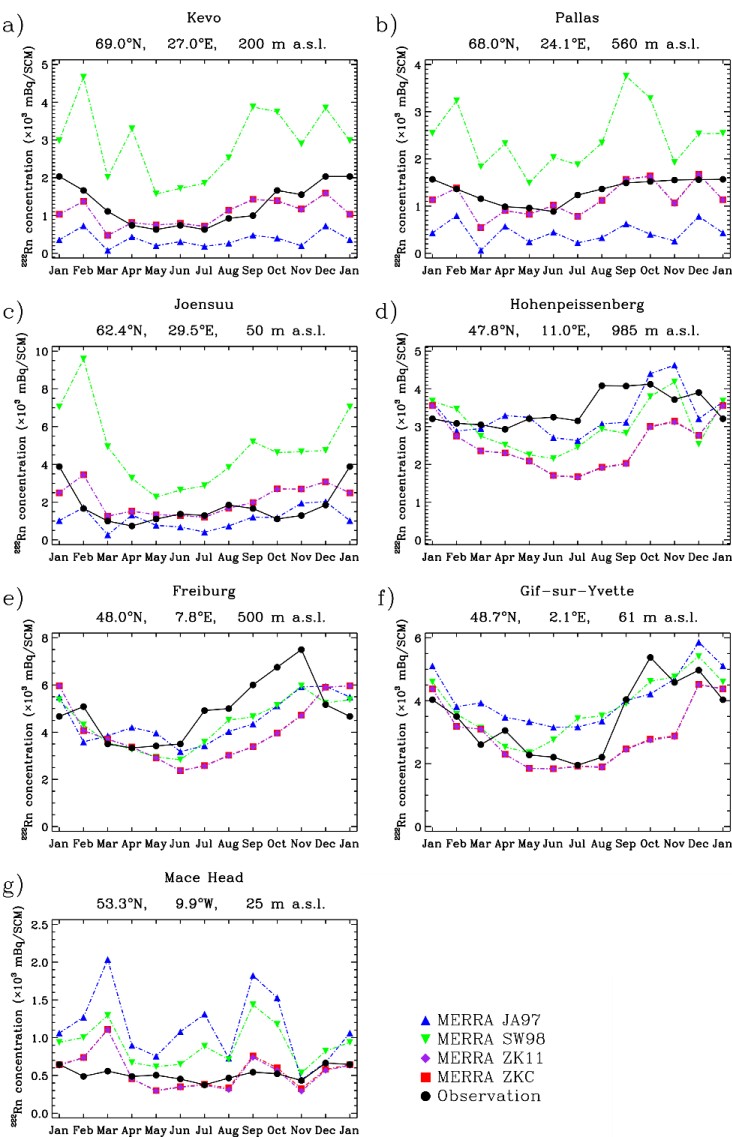

Figure 8. Comparison between simulated (color lines) and observed (black lines) monthly mean $^{222}$Rn concentrations (mBq/SCM) at selected surface sites in Europe. Location and elevation of each site are given upon each panel. See Table 1 for the list of model simulations. Note the small difference between the simulations with ZK11 and that with ZKC because of identical $^{222}$Rn emission in Europe.

Figure 9. Same as Fig. 8, but for Asia.





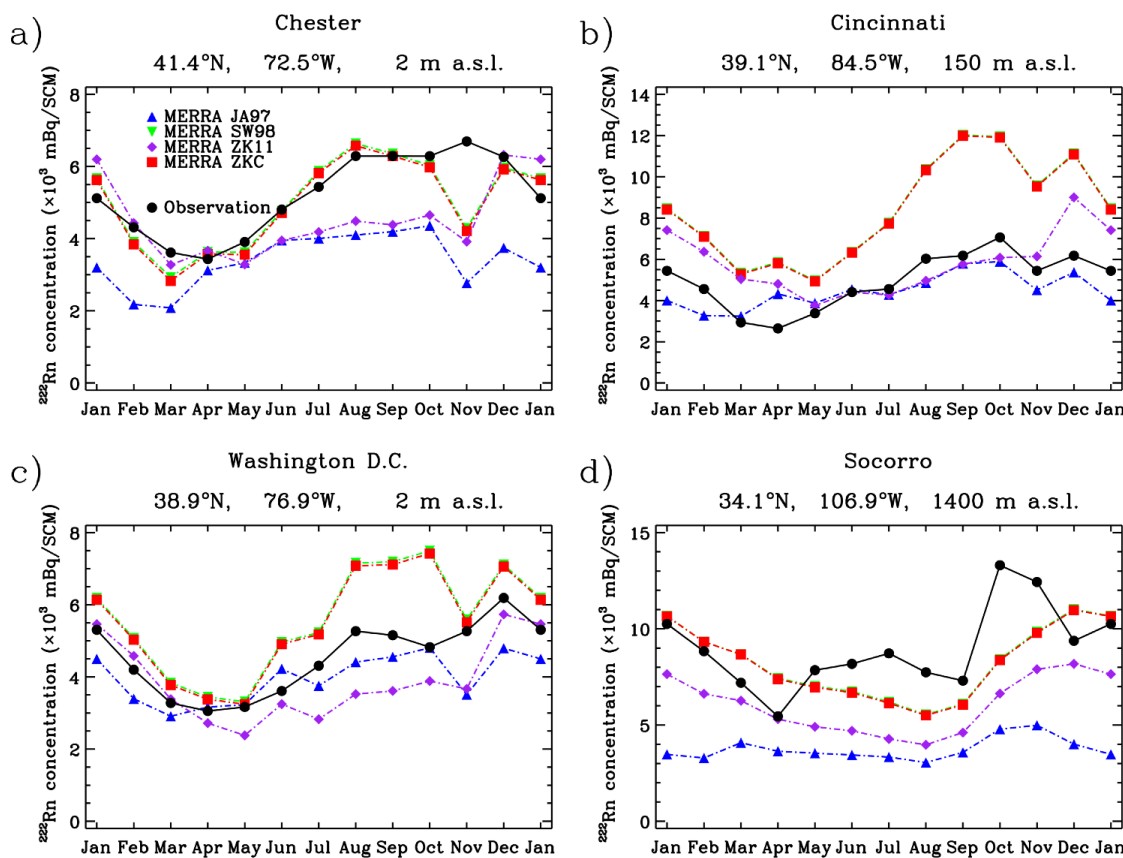

Figure 10. Same as Fig. 8, but for North America.



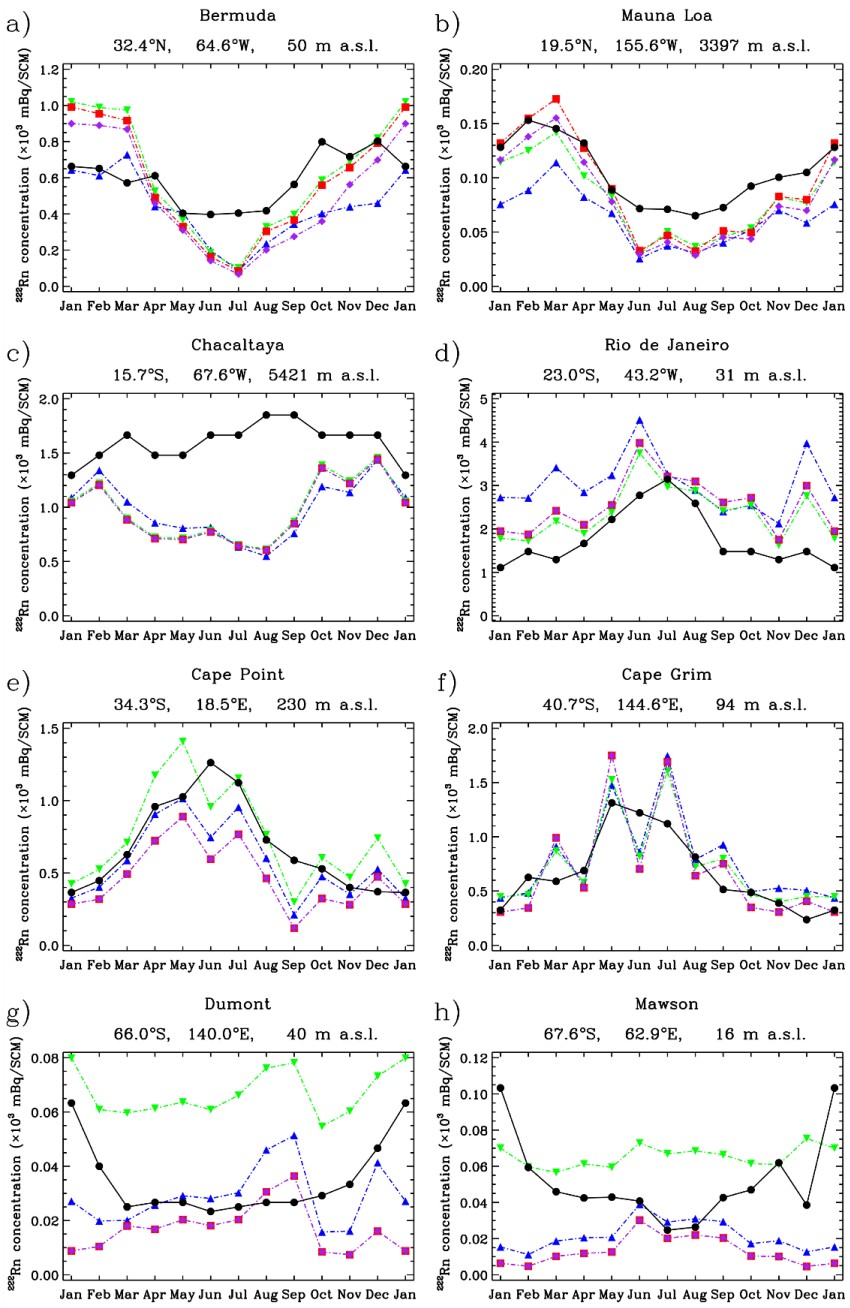

Figure 11. Same as Fig. 8, but for remote sites.



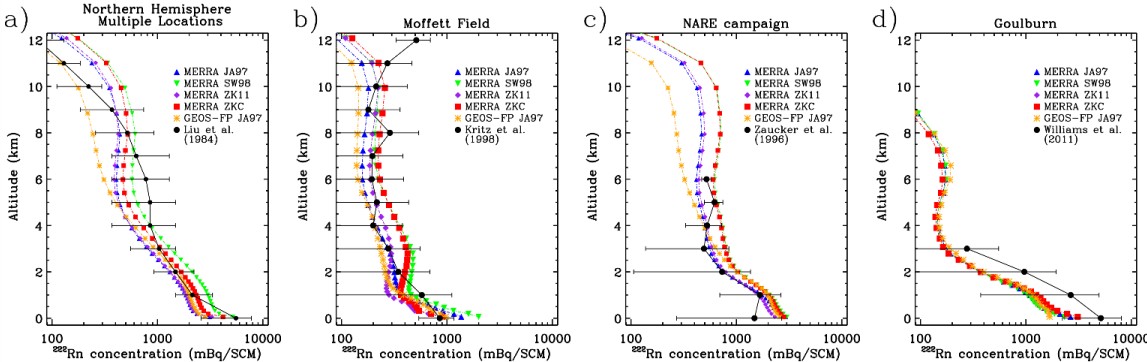

Figure 12. Comparison of vertical of $^{222}$Rn profiles (mBq/SCM) simulated with four emission scenarios (simulations A1, A2, A3, and A4, see Table 1) with a) an average profile compiled from multiple locations over the Northern Hemisphere continents (Liu et al., 1984), b) an average summertime profile constructed from measurements at Moffett Field (37.4°N, 122°W), California (Kritz et al., 1998), c) an average summertime profile from measurements on the east coast of Canada during the 1993 NARE campaign (Zaucker et al., 1996), and d) an averaged profile measured in May of 2006-2008, at Goulburn (34.8°S, 149.7°E), New South Wales, Australia (Williams et al., 2011). Horizontal bars indicated the standard deviations of the observed $^{222}$Rn concentrations.

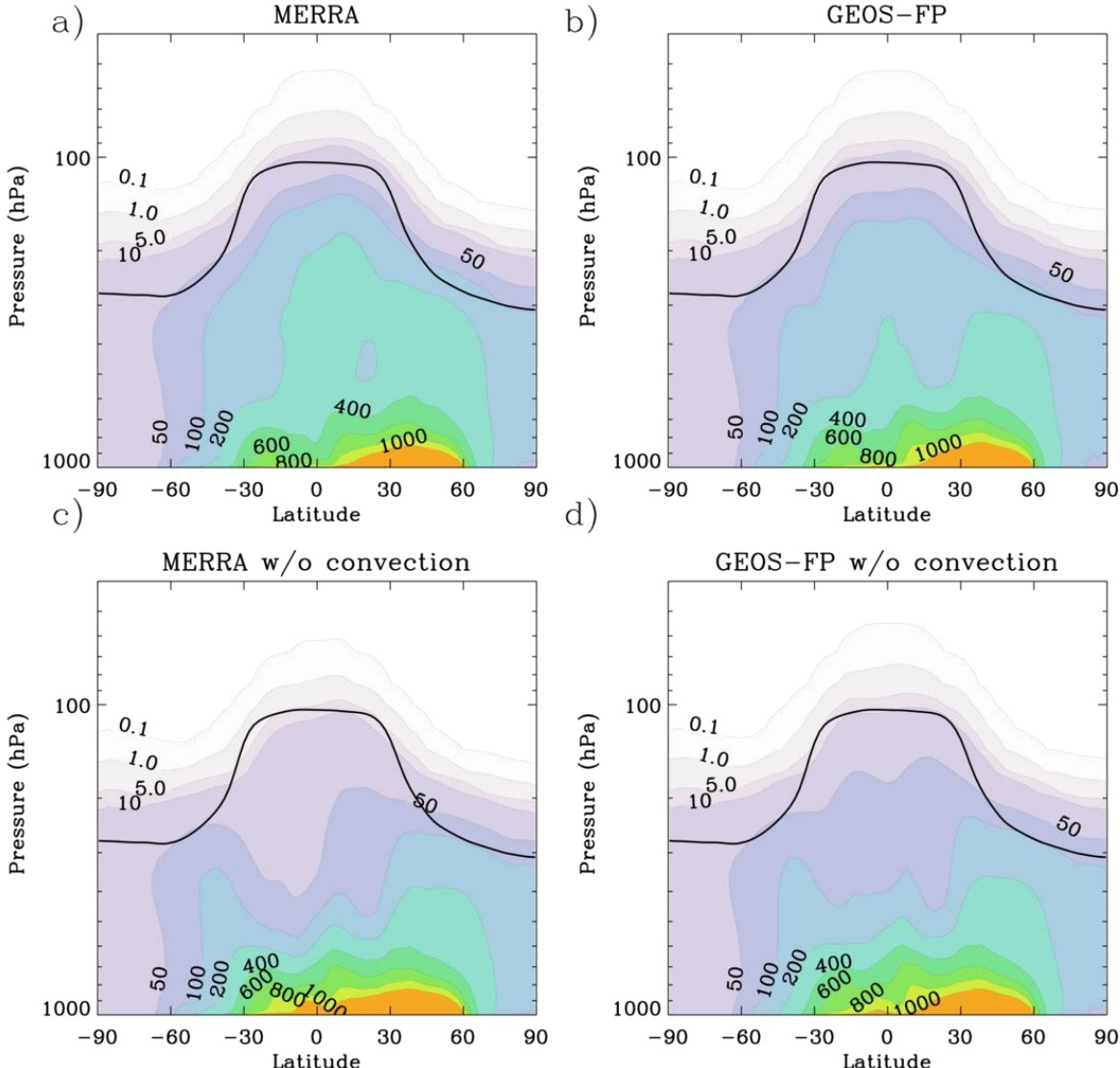

Figure 13. Latitude-pressure cross-sections of annual zonal mean $^{222}$Rn concentrations (mBq/SCM) as simulated by the GESO-Chem model driven by a) MERRA (A-1), b) GEOS-FP (B-1), c) MERRA without convection (A1-nc), and d) GEOS-FP without convection (B1-nc). Bold black lines denote the annual mean tropopause height (hPa) in the corresponding meteorological data set.





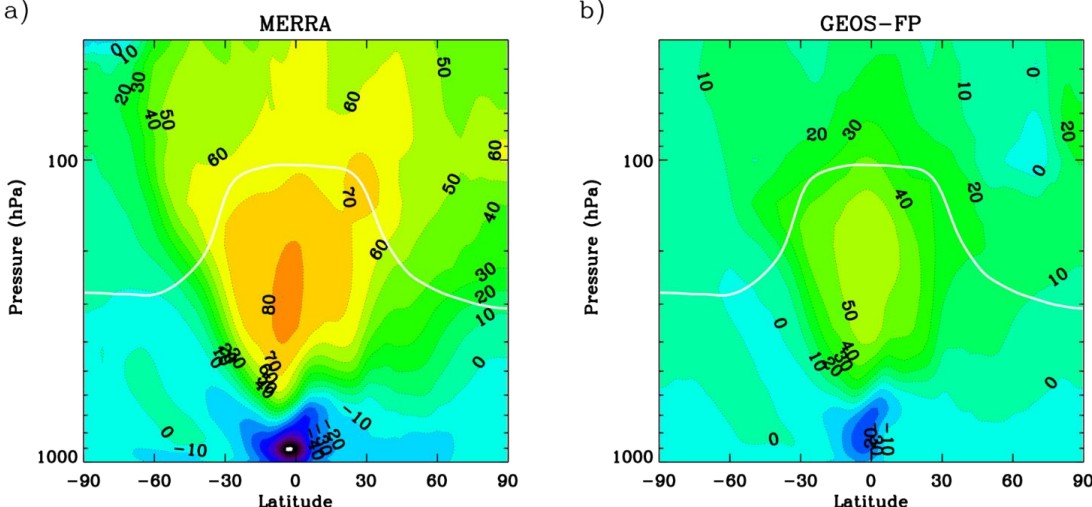

Figure 14. Percentages of annual zonal mean $^{222}$Rn concentrations contributed by convective transport in

5 the GEOS-Chem simulations driven by a) MERRA and b) GEOS-FP. Values are $(^{222}Rn - {}^{222}Rn_{nc})/^{222}Rn \times$

*100*, where $^{222}$Rn and $^{222}$Rn$_{nc}$ are $^{222}$Rn concentrations simulated with (A1 and B1, Table 1) and without

(A1-nc and B1-nc) the convection operator, respectively.

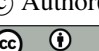

Figure 15. Comparison of annual zonal mean vertical fluxes of $^{222}$Rn ($\times10^{-22}$ kg m$^{-2}$ s$^{-1}$) in the GEOS-Chem simulations driven by MERRA and GEOS-FP. Upper panels: a) convective fluxes with MERRA, b) convective fluxes with GEOS-FP, and c) difference between a) and b). Lower panels: d) large-scale (LS) vertical fluxes with MERRA, e) large-scale vertical fluxes with GEOS-FP, and f) the difference between e) and f). The white lines indicate the tropopause height (hPa) in MERRA.



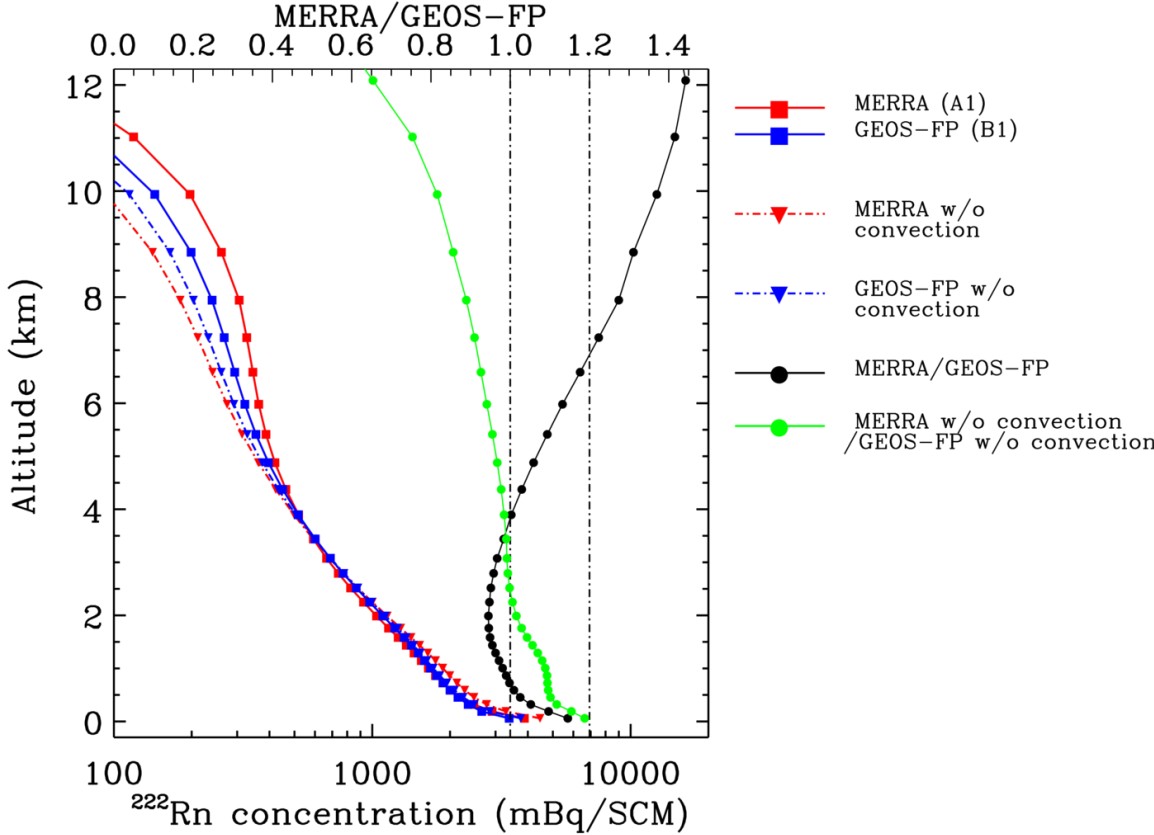

Figure 16. Annual zonal mean $^{222}$Rn profiles (mBq/SCM, red and blue lines) averaged over land areas
between 30-60°N latitudes in simulations driven by MERRA (A1 and A1-nc, Table 1) and GEOS-FP (B1
and B1-nc, Table 1), respectively. The black solid line (with the upper axis) shows the ratios of simulated
$^{222}$Rn concentrations in the standard simulations with MERRA and GEOS-FP. The green line shows the
same ratios when convection is turned off in the simulations. The two black dot-dashed lines have constant
ratios of 1.0 and 1.2, respectively.