# Peer review of "Emissions, seasonality, and convective transport"

_Atmospheric Chemistry and Physics, 2020_

## Referee Comment (RC1) · Anonymous Referee #1 · 2 Sep 2020

General Comments:

The sensitivity of Rn-222 to emission data sets is examined using GEOS-Chem driven by meteorological fields from two sources.

This manuscript would be stronger and of more scientific significance if it used MERRA-2 meteorological fields, which have been available for several years, and if the simulations were performed at $1 \times 1.25°$, which has become the standard resolution for global simulations.

The phrase "excessive Asian emissions" is confusing. It could be misinterpreted as indicating that model emissions are too high over Asia. Please go through manuscript

and rephrase as necessary.

Perhaps a Table could be added that concisely summarizes differences between the 4 emission scenarios. It's tedious reading through 4 pages describing the different emission scenarios

You correctly point out the deficiencies of using a simplistic emission inventory but you should also mention that there are benefits for using a simplistic emission inventory. For example, Rn-222 concentrations can be used as an indicator for how recently an air mass encountered land or a quick method of interpreting the effects of changes in the model configuration on convective mixing.

The method used to construct the ZKC inventory is a bit confusing. Does it consist of the SW98 inventory over North America and the ZK11 inventory elsewhere, with the latter multiplied by a factor of 1.2 over China? If yes, say this concisely and include the latitude/longitude range for the factor of 1.2 adjustment. Why is this adjustment applied to China and not southern China? If you want people to use this inventory rather than ZK11 you need to document it better – here.

In general, the figures are of excellent quality and enlightening.

Specific Comments P1 L25 (add lower bound to <70%)

P5 L7-8 Please rephrase awkward sentence that begins with "Due to the availability of"

P6 L19: TPCORE is internal nomenclature; please choose a more descriptive term such as monotonic if appropriate

P7: What is the difference between GEOS-FP and MERRA-2?

P9 L25: What is your rationale for not including these updates?

P10 L7 Define GCi

P12 L4: How many sites are located in the SH?

P12 L6: What do you mean by the Rn-222 observations were made in consecutive years. Is this true of all 51 sites? Do you mean at least 2 years of consecutive data are available at each site?

P12: Any thoughts on why Rn-222 profile data sets are scarce given the proliferation of other profile data sets?

P13: Would make for a more interesting read if "significantly", "substantially", and "remarkably" were quantified.

P16 L2: Why would the annual means be less representative? Were they obtained from only a portion of the year?

P19 L1-4: This is a bit confusing. Increasing the scaling factor over China from 1.2 to a higher value would lead to a better agreement between simulated and observed deposition fluxes over Asia (Figure 7c). However, this would also lead to an overestimate of deposition fluxes in the Northern Hemisphere. Please explain more clearly and/or reference the latter result.

P21 L3: Please quantify the model high-bias.

P25 L1: Are profiles of any more widely sampled trace gases useful for evaluating the convective detrainment level? I worry that some of the issue is with the observed profiles.

P30L15: Why would re-mapping have a greater impact on GEOS-FP than MERRA?

Figure 7b – be sure to indicate in caption that these are annual mean values of deposition fluxes. Also, be clear as to what each symbol shows and how many total there are. Is it 9 x 5?

Figure 8: Be clear that you are comparing an observed climatology from various years to a simulation of 2013. Are standard deviations of monthly means available at any of the sites? If yes, consider adding them.

Figure 12a. How much trust should we put into the observed profile based on multiple sites. How many of the profiles extended into the upper troposphere and were those from a small subset of the total locations?

Figure 13. When examining the effects of convection on trace gas profiles, it seems odd to show annual mean plots. Perhaps you should just show the summer hemisphere.

Figure 14. Please relabel this plot. The percent contribution cannot be zero. Perhaps call it the Percent Change in annual zonal mean 222-Rn due to convection

Technical Comments

P3 L10 shaping 222Rn –> shaping its

P3 L21 (remove period at end of line)

P3 L24 degree of discrepancies –> discrepancies

P6 L3: apparent changes –> changes

P7 L9: considerable –> a considerable

P11 L5 further low latitudes –> low latitudes

P11 L15: "changes are possible depending on the availability of measurements in these areas " –> changes are possible when measurements become available in these areas

P15L23: Replace excessive with large

P16 L19 Consider replacing "tentative" with "provisional"

P17 L11 three times at –> three times higher at

P29 L22 & L24: Replace excessive with very large

Figure 13: Replace GESO with GEOS.

[Figure]

P45 L8 : range with –> range within

---

## Referee Comment (RC2) · Anonymous Referee #2 · 17 Sep 2020

As far as I am aware of, this study presents the most comprehensive piece of work to date using 222Rn to evaluate atmospheric transport and mixing on a global scale. It includes the assessment of four 222Rn emission scenarios, a CTM driven by two meteorological data sets, and the comparison of simulations with practically all atmospheric 222Rn observations currently available, including vertical profiles. The clear structure of the paper, its great readability and meaningful displays make it a pleasure to read. It leaves no open question to me. There is very little that I can suggest to further improve it.

Minor comments

[Figure]

Differences between simulated and observed atmospheric concentrations occur for various reasons. One is the bias in measurement techniques, especially the underestimation of 222Rn concentrations derived from 222Rn progeny measurements near the surface (< 100 m above ground; cf. Grossi et al., 2020). Further, 222Rn concentration gradients within the first few metres above ground can be steep (e.g. Chambers et al., 2011). Several of the atmospheric observations in China were done between 1 and 1.5 m above ground (Jin et al., 1998, cited in Zhang et al, 2011, cited in the present study), which might explain some of the difference between simulation and observation for those sites. Are those sites represented in Figure 6 e-h by points indicating simulated values more than a factor of two smaller than observed values (or, better, observed values exceeding simulated values by more than a factor of two)?

Figure 6, y-axis label in the second row (Panel e) is "Observed ..." Should this not be "Simulated ...", as in the other rows?

Page 19, lines 7 and 8: "The seasonality in surface 222Rn concentrations is mainly affected by three factors: (1) the surface 222Rn emission flux rate determined by radium content and soil conditions; ..." This sentence is subject to eventual misinterpretation, in the way that radium content may be misunderstood as being seasonally variable. I would suggest to change the sentence to something like: "The seasonality in surface 222Rn concentrations is mainly affected by three factors: (1) seasonality in surface 222Rn emission flux rate resulting from seasonal changes in soil moisture, diffusivity, depth of the water table, snow and ice cover; ..."

Page 24, lines 1 and 2: Some 222Rn flux measurements from Antarctic soil are reported in Envangelista and Pereira (2002).

As mentioned in the text, there are vast regions without atmospheric 222Rn observations. Perhaps suggest, where from a modeller's perspective it would be desirable to see an atmospheric 222Rn detector established. Personally, I would very much like to see that happen at the tall tower (300 m) at Zotino (60° N 90 °E), in the middle of

[Figure]

Siberia (http://www.zottoproject.org/index.php/Main/Home).

References

Chambers et al. (2011) Separating remote fetch and local mixing influences on vertical radon measurements in the lower atmosphere. https://doi.org/10.1111/j.1600-0889.2011.00565.x

Grossi et al. (2020) Intercomparison study of atmospheric 222Rn and 222Rn progeny monitors. https://doi.org/10.5194/amt-13-2241-2020

Evangelista and Pereira (2002) Radon flux at King George Island, Antarctic Peninsula. https://doi.org/10.1016/S0265-931X(01)00137-0
* * *

---

## Author Comment (AC1) · 15 Dec 2020

We thank the reviewer for the thoughtful comments and suggestions. Our responses are in blue text. The line and page numbers in our responses refer to those in the revised manuscript with track changes.

General Comments:

The sensitivity of Rn-222 to emission data sets is examined using GEOS-Chem driven by meteorological fields from two sources.

This manuscript would be stronger and of more scientific significance if it used MERRA- 2 meteorological fields, which have been available for several years, and if the simulations were performed at $1 \times 1.25°$, which has become the standard resolution for global simulations.

We generally agree that our work would be more scientifically worthwhile if we could use the MERRA-2 reanalysis. However, MERRA was the most widely used reanalysis when we first started this study. To compare with transport in the more recent version of GEOS data, we included in the paper the model simulation driven by GEOS-FP ("Forward Processing"), the operational GEOS meteorological data product.

We also agree that a higher resolution would improve the simulation of transport in the model. However, the classic GEOS-Chem is not available at a resolution higher than $2 \times 2.5°$ for global simulations. The High-Performance version of GEOS-Chem (GCHP) runs at much higher resolution (c180, equivalent to $0.5 \times 0.625°$) with a cubed sphere grid geometry. Rn-222 and Pb-210 simulations with GCHP will be tested and evaluated in future studies.

The phrase "excessive Asian emissions" is confusing. It could be misinterpreted as indicating that model emissions are too high over Asia. Please go through manuscript and rephrase as necessary.

The phrase "excessive Asian emissions" was used in the manuscript to describe that the actual emissions in Asia are likely significantly higher than prescribed emissions in the model. We have replaced "excessive Asian emission" with "underestimated emission in Asia" in a few places to avoid confusion: Line 12 on Page 6, Line 14 on Page 10, Line 2 on Page 20, and Line 13 on Page 32.

Perhaps a Table could be added that concisely summarizes differences between the 4 emission scenarios. It's tedious reading through 4 pages describing the different emission scenarios. You correctly point out the deficiencies of using a simplistic emission inventory but you should also mention that there are benefits for using a simplistic emission inventory. For example, Rn-222 concentrations can be used as an indicator for how recently an air mass encountered land or a quick method of interpreting the effects of changes in the model configuration on convective mixing.

Thanks for the suggestion. We have added a new table (Table 1 in the revised manuscript and also given below) to summarize the differences between the four emission scenarios.

**Table 1. Global $^{222}$Rn emission scenarios used in this work.**

| Scenario | Reference | Description |
|---|---|---|
| JA97 | Jacob et al. (1997) | Emission fluxes are 1.0 atom cm$^{-2}$ s$^{-1}$ over land between 60°N – 60°S, 0.005 atom cm$^{-2}$ s$^{-1}$ between 60°N – 70°N and 60°S – 70°S, zero poleward of 70°N/S, and 0.005 atom cm$^{-2}$ s$^{-1}$ over lakes and oceans. Emissions are reduced by a factor of 3 when surface temperature is below 0°C. |
| SW98 | Schery and Wasiolek (1998) | Emission fluxes are formulated by using a theoretical diffusion model of porous soil with controlling factors of soil radium content, soil moisture, and surface temperature. Emission fluxes in SW98 were found to be overestimated and are reduced by a factor of 1.6 globally in this work (Koch et al., 2006; Zhang et al., 2011) |
| ZK11 | Zhang et al. (2011) | Based on SW98, ZK11 updated emission fluxes over Europe, U.S., China, Australia, and oceanic regions according to more recent measurements. |
| ZKC | This work | ZKC increases emission fluxes in the geographical territory of China by a factor of 1.2 upon ZK11 and retrogresses to SW98 over U.S. |

We agree with the reviewer that with a spatially uniform and simplistic emission, Rn-222 is a good tracer to assess continental influence and convective mixing. We have added the following statement in Line 15 on Page 8:

"Since the emission fluxes in JA97 are fairly uniform over land area, this simplistic emission scenario can be used to discern continental influence on air masses in global models and assess the effect of any changes in the model representation of convective mixing (Balkanski et al., 1992; Jacob et al., 1997)."

Our perspective about the value of using a realistic Rn-222 emission scenario is that model simulated Rn-222 concentrations can be better compared with actual observations, thus providing an observation-based evaluation of transport processes in models.

The method used to construct the ZKC inventory is a bit confusing. Does it consist of the SW98 inventory over North America and the ZK11 inventory elsewhere, with the latter multiplied by a factor of 1.2 over China? If yes, say this concisely and include the latitude/longitude range for the factor of 1.2 adjustment. Why is this adjustment applied to China and not southern China? If you want people to use this inventory rather than ZK11 you need to document it better – here.

A description of the modifications we made to the ZKC inventory is now given in the added Table 1 (also see above). The factor of 1.2 adjustment was applied to the geographical territory of China.

The adjustment of Rn-222 emission fluxes over China (not just southern China) is made based on the analyses of Rn-222 surface concentrations and Pb-210 deposition fluxes (Du et al., 2015). Some sites used in the Pb-210 deposition analysis are located in northern and inland China. We have clarified this in Line 6 on Page 20:

"We then calculate the correlations ... at the sites in North America (nine sites) and Asia (nine sites; Du et al. (2015)). Some studied Asian sites are located in northern and inland China."

In general, the figures are of excellent quality and enlightening.

**Specific Comments**

P1 L25 (add lower bound to <70%)

There is no lower bound. The "<70%" is an approximate fraction of data within the a-factor-of-2 range. To avoid confusion, we have replaced "<70%" by the exact value 68.9% in Fig. 6(m).

P5 L7-8 Please rephrase awkward sentence that begins with "Due to the availability of"

The original sentence reads:

"Due to the availability of extensive measurements of $^{222}$Rn emission fluxes and surface concentrations, Europe has the finest resolution emission inventory of up to 0.083° ×0.083° with variability in regional and temporal emissions"

We have rewritten the sentence as the following:

"Published $^{222}$Rn emission inventories for Europe have very fine spatial resolutions (up to 0.083° ×0.083°) with monthly variability due to extensive measurements of emission fluxes and surface concentrations across the continent."

P6 L19: TPCORE is internal nomenclature; please choose a more descriptive term such as monotonic if appropriate.

We have rewritten the sentence as the following:

"The model uses a flux-form semi-Lagrangian finite volume scheme, known as TPCORE, to calculate advection (Lin and Rood, 1996). The scheme uses the monotonic piecewise parabolic method under convergence conditions and a semi-Lagrangian method otherwise."

P7: What is the difference between GEOS-FP and MERRA-2?

GEOS-FP is the current operational product of GEOS-5 and includes recent developments of the model. MERRA and MERRA-2 are long-term reanalysis products based on different versions of GEOS-5. We have revised the relevant sentence to "MERRA-2, which is based on a newer version of GEOS-5 and shows improved climate over MERRA (Molod et al., 2015), is not used here ….".

P9 L25: What is your rationale for not including these updates?

Zhang et al. (2011) showed that Rn-222 emissions in EU calculated based on gamma-dose rate (Szegvary et al., 2009) lead to satisfactory agreement between model simulated and observed Rn-222 surface concentrations. Both López-Coto et al. (2013) and Karstens et al. (2015) used a different type of method. They used soil conditions to estimate Rn-222 emission fluxes and the uncertainties are largely determined by soil moisture in land surface models. Considering our second goal of assessing

convection in GEOS-Chem and a better comparison with the results in Zhang et al. (2011), we chose to follow the method and analysis in Zhang et al. (2011). We have revised the statement in Line 16 on Page 10 to the following:

"Since ZK11 has been tested with satisfactory agreements between modeled and observed surface concentrations in Europe (Zhang et al., 2011), the updates for emission fluxes in Europe by López-Coto et al. (2013) and Karstens et al. (2015) are not included."

P10 L7 Define Gucci

Added in the text "GCi (Giga-Curie)".

P12 L4: How many sites are located in the SH?

Added in the text "Fewer sites (11) are located in the Southern Hemisphere".

P12 L6: What do you mean by the Rn-222 observations were made in consecutive years. Is this true of all 51 sites? Do you mean at least 2 years of consecutive data are available at each site?

We have changed "consecutive years" to "multiple years", as the measurements were not always consecutive.

P12: Any thoughts on why Rn-222 profile data sets are scarce given the proliferation of other profile data sets?

Thanks for bringing this up. We have added one sentence in the text: "The scarcity of $^{222}$Rn airborne measurements is partly due to the fact that the measurement requires an extraction and counting facility nearby in order to minimize decay and that the process of radon extraction is labor-intensive (Williams et al., 2011)."

P13: Would make for a more interesting read if "significantly", "substantially", and "remarkably" were quantified.

We have quantified the changes in surface concentration differences in Line 12-13 and in Line 22-23 on Page 14.

P16 L2: Why would the annual means be less representative? Were they obtained from only a portion of the year?

Some annual means at China sites were obtained from measurements over the period of Nov. 1988 - Jan. 1990. The measurements only covered a little over a year and were less representative for the surface climatology compared to other sites where monthly measurements were reported for multiple years. We have added the following in the earlier data section:

"The few inland sites in China only reported annual means based on measurements of 1-2 years (Jin et al., 1998)."

Jin, Y., Iida, T., Wang, Z., Ikebe, Y., and Abe, S.: A subnationwide survey of outdoor and indoor 222Rn concentrations in China by passive method. Radon and thoron in the human environment, in: Radon and Thorn in the Human Environment, in: Proceedings of the 7th Tohwa University International

Symposium, edited by: Katase, A. and Shimo, M., World Scientific Publishing Co. Pre. Ltd., Singapore, 276–281, 1998.

P19 L1-4: This is a bit confusing. Increasing the scaling factor over China from 1.2 to a higher value would lead to a better agreement between simulated and observed deposition fluxes over Asia (Figure 7c). However, this would also lead to an overestimate of deposition fluxes in the Northern Hemisphere. Please explain more clearly and/or reference the latter result.

We are working on a follow-up manuscript to compare model results with Pb-210 observations, including deposition fluxes and vertical profiles. The unpublished results indicate that larger scaling factors will lead to large overestimates over the rest of the Northern Hemisphere. We have changed the sentence to "…we choose to use a moderate scaling factor of only 1.2 for China to avoid large overestimates of total $^{210}$Pb deposition fluxes over the rest of the Northern Hemisphere". The reference of the on-going Pb-210 work has been given in Line 4-5 on Page 17.

P21 L3: Please quantify the model high-bias.

We have added quantifications of model high biases here:

"Simulations with JA97 and SW98 overestimate the observations by a factor of >2 on average, while such large overestimates are only seen in February for ZK11 and ZKC."

P25 L1: Are profiles of any more widely sampled trace gases useful for evaluating the convective detrainment level? I worry that some of the issue is with the observed profiles.

Vertical distribution of CO has been frequently used to examine convective transport in atmospheric models (e.g., Allen et al., 1996; Ott et al., 2009). Stanfield et al. (2019) found that the frequency distribution of the convective entrainment rates (mixing between environmental air with in-cloud air) for deep convection events in GEOS-5 has a significantly larger fraction in the higher-end values compared to the rates derived from TES/MLS-observed CO profiles. Intensive mixing within convective updraft undermines the upward lifting of surface air masses to the upper troposphere, possibly causing the rapidly decreasing $^{222}$Rn concentrations with height in the simulation with GEOS-FP (Figure 12a). The cloud-top height for convective clouds in MERRA is likely biased high according to a comparison with CERES-observed clouds (Posselt et al., 2012). These are consistent with our conclusions about the detrainment level derived based on $^{222}$Rn profiles. We have revised this part of the paragraph to the following to support our viewpoint:

"... MERRA exhibits a higher and deeper convection from 5 to 10 km. As a result, a remarkable underestimation of $^{222}$Rn concentrations with MERRA is seen from 4 to 8 km, followed by overestimations above 9 km. Deep convective cloud top in MERRA has been shown biased high compared to CERES-observed clouds (Posselt et al., 2012). Stanfield et al. (2019) found that the frequency distribution of convective entrainment rates (mixing between environmental air with in-cloud air) for deep convection events in GEOS-5 has a significantly larger fraction in the higher-end values compared to the rates derived from TES/MLS-observed CO profiles. Intensive mixing during convective updraft undermines the upward lifting of surface air masses to the upper troposphere, possibly causing the rapidly decreasing $^{222}$Rn concentrations with height in the simulation with GEOS-FP. Due to weaker

convection in GEOS-FP, the simulation underestimates in a broader altitude range (4-10 km). It seems challenging for the two GEOS products to capture the convective detrainment level. ..."

Allen, D. J., Kasibhatla, P., Thompson, A. M., Rood, R. B., Doddridge, B. G., Pickering, K. E., ... & Lin, S. J. (1996). Transport-induced interannual variability of carbon monoxide determined using a chemistry and transport model. *Journal of Geophysical Research: Atmospheres*, *101*(D22), 28655-28669.

Ott, L. E., Bacmeister, J., Pawson, S., Pickering, K., Stenchikov, G., Suarez, M., ... & Xueref-Remy, I. (2009). Analysis of convective transport and parameter sensitivity in a single column version of the Goddard earth observation system, version 5, general circulation model. *Journal of the atmospheric sciences*, *66*(3), 627-646.

Posselt, D. J., Jongeward, A. R., Hsu, C. Y., & Potter, G. L. (2012). Object-based evaluation of MERRA cloud physical properties and radiative fluxes during the 1998 El Niño–La Niña transition. *Journal of climate*, *25*(21), 7313-7327.

Stanfield, R. E., Su, H., Jiang, J. H., Freitas, S. R., Molod, A. M., Luo, Z. J., ... & Luo, M. (2019). Convective entrainment rates estimated from Aura CO and CloudSat/CALIPSO observations and comparison with GEOS-5. *Journal of Geophysical Research: Atmospheres*, *124*(17-18), 9796-9807.

P30L15: Why would re-mapping have a greater impact on GEOS-FP than MERRA?

Re-mapping (from the cubed-sphere to equally rectilinear grids) itself does not have different impacts on GEOS-FP than MERRA. Convection in the simulation driven by GEOS-FP is affected more by a more intensive regridding from a finer native model resolution (0.25° by 0.3125° to 2° by 2.5°) compared to MERRA (0.5° by 0.667° to 2° by 2.5°). We added such information in an earlier place in Line 24 on Page 28:

"GEOS-FP has a finer native horizontal resolution (0.25° latitude by 0.3125° longitude) than MERRA reanalysis (0.5° latitude by 0.667° longitude), and exhibits weaker convection likely due to a more intensive regridding."

We also revised the sentence in Line 5 on Page 33 to "The weak convection in GEOS-FP leads to large low biases of $^{222}$Rn in the mid-high troposphere".

Figure 7b – be sure to indicate in caption that these are annual mean values of deposition fluxes. Also, be clear as to what each symbol shows and how many total there are. Is it 9 x 5?

Now we indicate in the caption that the values of deposition fluxes are annual means. The number of model simulations (5) and surface sites (9) have been stated in the figure caption already. There are 9 x 5 = 45 points in total.

Figure 8: Be clear that you are comparing an observed climatology from various years to a simulation of 2013. Are standard deviations of monthly means available at any of the sites? If yes, consider adding them.

We have changed the first sentence in the caption to:

"Comparison between observed [222]Rn climatological monthly means (black lines) and simulated monthly means in 2013 (color lines) at selected surface sites in Europe."

Unfortunately, standard deviations are available for only a few sites. For those sites, multiple years of monthly data are available from the data repository submitted with this manuscript.

Figure 12a. How much trust should we put into the observed profile based on multiple sites. How many of the profiles extended into the upper troposphere and were those from a small subset of the total locations?

As stated earlier in the manuscript, the composite profile is compiled from summertime observations at 23 sites over the Northern Hemisphere mid-latitude continental regions. It should provide a decent measure of summertime vertical [222]Rn distribution over land. More than half of the profiles reach up to 6-12 km; the composite profile is thus not biased by a small subset of observations in the upper troposphere. Now we state in the Fig. 12 caption "….from 23 locations over the Northern Hemisphere continents (Liu et al., 1984),….. In panel (a), more than half of the observed profiles reach up to 6-12km".

Figure 13. When examining the effects of convection on trace gas profiles, it seems odd to show annual mean plots. Perhaps you should just show the summer hemisphere.

This comment is addressed together with the next one.

Figure 14. Please relabel this plot. The percent contribution cannot be zero. Perhaps call it the Percent Change in annual zonal mean 222-Rn due to convection

We agree with the suggestion that summertime results better reflect the effects of convection. We have replaced annual zonal means in Figures 13&14 with zonal means averaged over June-July-August. The conclusions from these analyses are still the same except that the effects of convection are more obvious in the Northern Hemisphere. The relevant pieces of text have been updated accordingly. The figure 14 caption has been revised to "Percentage changes in zonal mean [222]Rn concentrations averaged over June-July-August due to convective transport in the GEOS-Chem simulations …." (see below).

[Figure]

Figure 13. Latitude-pressure cross-sections of zonal mean [222]Rn concentrations averaged over June-July-August (mBq/SCM) as simulated by the GEOS-Chem model driven by a) MERRA (A-1), b) GEOS-FP (B-1), c) MERRA without convection (A1-nc), and d) GEOS-FP without convection (B1-nc). Bold black lines denote the zonal mean tropopause height (hPa) in the corresponding meteorological data set.

[Figure]

Figure 14. Percentage changes in zonal mean [222]Rn concentrations averaged over June-July-August due to convective transport in the GEOS-Chem simulations driven by a) MERRA and b) GEOS-FP. Values

are $(^{222}Rn - {}^{222}Rn_{nc})/^{222}Rn \times 100$, where $^{222}Rn$ and $^{222}Rn_{nc}$ are $^{222}Rn$ concentrations simulated with (A1 and B1, Table 1) and without (A1-nc and B1-nc) the convection operator, respectively.

**Technical Comments**

P3 L10 shaping 222Rn –> shaping its

Done.

P3 L21 (remove period at end of line)

It is a comma not period. We will keep it there.

P3 L24 degree of discrepancies –> discrepancies

P6 L3: apparent changes –> changes

P7 L9: considerable –> a considerable

P11 L5 further low latitudes –> low latitudes

All done.

P11 L15: "changes are possible depending on the availability of measurements in these areas " –> changes are possible when measurements become available in these areas

P15L23: Replace excessive with large

P16 L19 Consider replacing "tentative" with "provisional"

P17 L11 three times at –> three times higher at

All done.

P29 L22 & L24: Replace excessive with very large

This has been addressed in the second major comment above.

Figure 13: Replace GESO with GEOS.

P45 L8: range with –> range within

Done.

---

## Author Comment (AC2) · 15 Dec 2020

We thank the reviewer for the comments and insights. Our responses are in blue text. The line and page numbers in our responses refer to those in the revised manuscript with track changes.

As far as I am aware of, this study presents the most comprehensive piece of work to date using 222Rn to evaluate atmospheric transport and mixing on a global scale.  It includes the assessment of four 222Rn emission scenarios, a CTM driven by two meteorological data sets, and the comparison of simulations with practically all atmospheric222Rn observations currently available, including vertical profiles.  The clear structure of the paper, its great readability and meaningful displays make it a pleasure to read. It leaves no open question to me. There is very little that I can suggest to further improve it.

Minor comments

Differences between simulated and observed atmospheric concentrations occur for various reasons. One is the bias in measurement techniques, especially the underestimation of 222Rn concentrations derived from 222Rn progeny measurements near the surface (< 100 m above ground; cf. Grossi et al., 2020). Further, 222Rn concentration gradients within the first few metres above ground can be steep (e.g. Chambers et al., 2011). Several of the atmospheric observations in China were done between 1 and 1.5 m above ground (Jin et al., 1998, cited in Zhang et al, 2011, cited in the present study), which might explain some of the difference between simulation and observation for those sites. Are those sites represented in Figure 6 e-h by points indicating simulated values more than a factor of two smaller than observed values (or, better, observed values exceeding simulated values by more than a factor of two)?

The comparisons between observations and model results for Asian sites (Fig. 6e-h) suggest that surface $^{222}$Rn concentrations were underestimated by at least a factor of two in the model for a few sites. As the reviewer pointed out, the measurements have been taken very close to the surface. According to Figure 1 in Chambers at al. (2011), $^{222}$Rn concentrations measured close to surface can be significantly higher than those at 50m between 8pm until 9am, and are possibly higher than the average taken from the model bottom-layer gridbox (~100m high). On the other hand, there are possible low biases in the measurements due to measurement techniques. Such low biases may partially compensate for the underestimate due to the steep concentration gradients near the surface. Considering these rationales, we have added the following discussion in Line 16 on Page 16:

"The observations in China were taken between 1m and 1.5m above ground according to Jin et al. (1998). The model surface layer concentrations usually represent the averages in the model bottom layer (~100m high), and thus may be literally lower than the observations due to the steep concentration gradients near the surface, especially during nighttime (Chambers et al., 2011). On the other hand, there are possible low biases in the $^{222}$Rn concentrations derived from $^{222}$Rn progeny measurements (Schmithüsen et al., 2017; Grossi et al., 2020), lessening the above model underestimate due to large near-surface vertical gradients. These biases differ on a case-by-case basis and are difficult to quantify."

Figure 6, y-axis label in the second row (Panel e) is "Observed ..." Should this not be "Simulated ...", as in the other rows?

Thanks for catching this typo. Now corrected.

Page 19, lines 7 and 8: "The seasonality in surface 222Rn concentrations is mainly affected by three factors: (1) the surface 222Rn emission flux rate determined by radium content and soil conditions; ..." This sentence is subject to eventual misinterpretation, in the way that radium content may be misunderstood as being seasonally variable. I would suggest to change the sentence to something like: "The seasonality in surface 222Rn concentrations is mainly affected by three factors: (1) seasonality in surface 222Rn emission flux rate resulting from seasonal changes in soil moisture, diffusivity, depth of the water table, snow and ice cover; ..."

Thanks for the suggestion. We have revised the sentence to:

"The seasonality in surface $^{222}$Rn concentrations is mainly affected by three factors: (1) the variability in $^{222}$Rn emission flux rate due to seasonal changes in soil moisture, diffusivity, depth of the water table, and snow and ice coverage; ..."

Page 24, lines 1 and 2: Some 222Rn flux measurements from Antarctic soil are reported in Envangelista and Pereira (2002). As mentioned in the text, there are vast regions without atmospheric 222Rn observations. Perhaps suggest, where from a modeller's perspective it would be desirable to see an atmospheric 222Rn detector established. Personally, I would very much like to see that happen at the tall tower (300 m) at Zotino (60 N 90 E), in the middle of Siberia (http://www.zottoproject.org/index.php/Main/Home).

We thank the reviewer for pointing us to this $^{222}$Rn flux measurement work in the Antarctic. The work provides valuable measurement of $^{222}$Rn fluxes during the summer of 1998/1999 at the Admiralty Bay area of King George Island, Antarctic Peninsula (62°S, 58°W). Reported fluxes ranged between $0.21 \times 10^{-2}$ atom cm$^{-2}$ s$^{-1}$ and $28 \times 10^{-2}$ atom cm$^{-2}$ s$^{-1}$. We have added the following discussion in the text:

"Evangelista and Pereira (2002) reported summertime $^{222}$Rn fluxes ranging between $0.21 \times 10^{-2}$ atom cm$^{-2}$ s$^{-1}$ and $28 \times 10^{-2}$ atom cm$^{-2}$ s$^{-1}$ during the summer of 1998/1999 at the Admiralty Bay area of King George Island, Antarctic Peninsula (62°S, 58°W). The work also suggested such low fluxes could not explain $^{222}$Rn concentration surges in the atmosphere. The sparse measurements at the edge of the Antarctic are not adequate for inferring emission fluxes over the remote continent."

Regarding the desire for more observations, we have made a few statements in section 3.2 to suggest more measurements in Asia, North America, and Antarctic. Measurements in the middle Siberia would be very valuable because they may help quantify $^{222}$Rn emissions and surface concentrations in the northern Asia.

References

Chambers et al. (2011) Separating remote fetch and local mixing influences on vertical radon measurements in the lower atmosphere. https://doi.org/10.1111/j.1600- 0889.2011.00565.x

Grossi et al. (2020) Intercomparison study of atmospheric 222Rn and 222Rn progeny monitors. https://doi.org/10.5194/amt-13-2241-2020

Evangelista and Pereira (2002) Radon flux at King George Island, Antarctic Peninsula. https://doi.org/10.1016/S0265-931X(01)00137-0

---

## Author Comment (AC3) · 15 Dec 2020

We thank the editor for his questions. Our replies are in blue text.

There are two questions for which I did not find the answer in the manuscript:

1- What is the effect of the model resolution, i.e. would you get better agreement (or not?) with a 0.1°x0.1° horizontal resolution?

As stated in our response to Review#1, we do think a higher resolution would improve the simulation of transport (i.e., minimizing lost vertical transport due to the regridding of meteorological fields) in the model. In the context of Rn-222 modeling, we doubt that model simulations with very fine resolutions can improve the scientific results of this work, largely due to the uncertainties and coarse resolution associated with Rn-222 emission maps. Studies of Rn-222 emission fluxes so far do not provide sufficient data required to produce fine-resolution emission maps, except for Europe. On the other hand, the classic version of the GEOS-Chem model is not available for conducting global simulations at a resolution higher than $2° \times 2.5°$. In future studies, Rn-222/Pb-210 simulations will be tested with GCHP, the High-Performance version of GEOS-Chem that can be run at high resolution with a cubed sphere geometry.

2- What is the effect of the geographical position of the 222Rn station in the model grid box when the station is on the coast such as for Fuzhou? Have you thought what would change in your simulated concentration if the model box was 80% marine / 20% continental versus 80% continental / 20% marine?

We appreciate Editor's insight that the observations and sampled model results may represent different air masses at coastal regions due to the nonuniform surface type in the model grid box. For coastal sites such as Fuzhou, the Rn-222 observations could be largely affected by local emissions and thus more typical of those at inland sites. Coastal sites are usually positioned within model gridboxes that are partly land and partly water. Some of these gridboxes may be dominated by water whereas the sites could be located near the edges of the gridboxes and not close to the water. In such cases, while the land area fractions (excluding sea and lake areas) are taken into account in the model calculation of Rn-222 emission fluxes for these gridboxes, the model-simulated concentrations may not represent the observed at the sites due to the subgrid nature of the latter.

We thus tried sampling the model results at adjacent gridboxes and found that the model results for the gridbox to the west of the original (moving away from coast) are much more comparable to the observed magnitude and seasonality of surface Rn-222 concentrations at Fuzhou. This suggests that the observations are significantly affected by local Rn-222 emissions. We also found a similar improvement at the site of Hong Kong. We replaced model results in Figure 9(c,d) with these better comparisons and moved the original figure panels to the supplementary materials.

The following has been added in the text (Page 23 in the revised manuscript with track changes): "At two China coastal sites, Fuzhou and Hong Kong, the model results at the corresponding grid boxes are much lower than the observations (Fig. S4). We tried sampling the model results at

adjacent gridboxes and found that those for the gridbox to the west are much more comparable to the observed (Fig. 9c and 9d). This suggests that the observations at both sites are significantly affected by local $^{222}$Rn emissions. The $^{222}$Rn observations show a minimum in summer, reflecting the intrusion of low-$^{222}$Rn marine air associated with the Asian summer monsoon. Although the model successfully captures the observed seasonality, the simulation with ZKC (with enhanced emissions in China) shows a much better agreement compared to the large low bias in the simulation with JA97."

Revised Figure 9:

[Figure]

Figure 9. Same as Fig. 8, but for Asia. Note that the model results used in panel c) Fuzhou and d) Hong Kong are sampled at the gridboxes to the west of the ones where the sites are located to achieve a better agreement with the observations. See text for details.

In the supplementary materials, we added:

[Figure]

Figure S4. Same as Fig. 9 (c) and (d), but the simulated $^{222}$Rn concentrations are sampled at the model gridboxes corresponding to the site locations.